# An Early Warning System for Currency Crises in Emerging Countries

**Lutfa Tilat Ferdous** [1,*] **, Khnd Md Mostafa Kamal** [2] **, Amirul Ahsan** [3] **, Nhung Hong Thuy Hoang** [4] **and Munshi Samaduzzaman** [5]

[1] Department of Economics, Finance and Accounting, University of Leicester, Leicester LE1 7RH, UK
[2] Department of Information Systems and Business Analytics, Deakin University, Melbourne, VIC 3217, Australia; m.kamal@deakin.edu.au
[3] Department of Finance, Deakin University, Melbourne, VIC 3217, Australia; amirul@deakin.edu.au
[4] School of Economics, University of Economics Ho Chi Minh City, Ho Chi Minh 722700, Vietnam; nhung.hth@vpn.edu.vn
[5] School of Business and Law, Central Queensland University, Brisbane, QLD 4701, Australia; m.samaduzzaman@cqu.edu.au
[*] Correspondence: ltf4@leicester.ac.uk

**Abstract:** In this study we develop an early warning system (EWS) to forecast currency crises in emerging countries in Asia and Latin America, using logit regression on monthly data from 1992 to 2011. We found that macroeconomic and institutional variables are valuable indicators for forecasting crises. Our results show that a low level of export growth, current account surplus/GDP, GDP growth, a high level of real exchange rate growth, import growth, and short-term debt/reserves can explain the advent of a possible currency crisis. We found that a poor law and order scenario and high external conflict can lead to a currency crisis. Additional findings include high government stability and the absence of internal conflict, which contribute to an absence of democracy, ultimately leading to a currency crisis. The policy-makers can consider taking the effective pre-emptive actions to prevent the currency crises occurring in the future.

**Keywords:** currency crisis; early warning system; emerging countries; logit model

## 1. Introduction

We develop an early warning system (EWS) model to forecast currency crises in emerging countries. There were numerous financial crises during the 1990s, these include in Europe from 1992 to 1993; Mexico from 1994 to 1995; Asia from 1997 to 1998; Brazil in 1999; Turkey in 2001; Argentina in 2002; and the global financial crisis (GFC), originating in the USA in 2007. These financial crises greatly affected the countries' economic systems, politics, and society. They caused huge uncertainty and anxiety, resulting from high inflation, slow growth, high unemployment, and poverty. Subsequently, GDP growth rates declined, and this caused changes in nominal exchange rates, coupled with high devaluation. In Argentina, the GDP dropped by 20%, so real wages further declined by a similar margin. The cost of any crisis is very high for any economy. Consequently, policymakers are constantly under pressure to seek new policies to save the affected economies. Having frequent crises has led to increased investigations, for the construction of monitoring tools that can forecast crises in advance. These studies examined what is referred to as an early warning system (EWS).

There are three common types of financial crises: currency, banking, and debt. However, this study focuses on currency crisis only. EWS models for currency crises were introduced by Krugman (1979) and further enhanced by Flood and Garber (1984) and Obstfeld (1995, 1996), who proposed a different model for forecasting currency crises. He stated that currency crises occurred due to over-speculation. However, the model fails to

take time variables into account. After the Asian Financial Crisis of 1997–1998, Kaminsky and Reinhart (1999) built a model for twin crises that combines banking and currency crises. This research indicated that a banking crisis often occurred just prior to a currency crisis, causing it to happen, this deepened the currency crisis, and, consequently, the economy entered a twin crisis. These studies applied macroeconomic and financial indicators to measure currency crises (Kaminsky et al. 1998; Frankel and Rose 1996; Berg and Pattillo 1999a). In recent years, some researchers have identified institutional variables and applied them to forecast the probability of imminent crises (Shimpalee and Breuer 2006; Leblang and Satyanath 2008). Previously, these models have employed a variety of indicators to measure currency crises. One study emphasizes a set of indicators, whereas another analysis prefers a different set of variables. The following section assesses these previous studies and highlights the need for a robust EWS model. This study develops such a model, taking all necessary indicators together, for effective and comprehensive currency crisis modelling.

The optimal cut-off threshold is an essential part of identifying a currency crisis effectively. Different methods are applied to determine the optimal threshold. However, not all of these need to apply the optimal cut-off threshold in an EWS model. Kaminsky et al. (1998) developed a noise-signal-ratio (NSR) model to find the desired threshold to minimize the ratio of false signals, compared to correct signals. Berg and Pattillo (1999b) applied the quadratic probability score (QPS) and the log probability score (LPS) for a better forecasting ability than the noise-signal-ratio (NSR). Bussiere and Fratzscher (2006) built a loss function model, to help policymakers forecast currency crises, applying the Demirgüç-Kunt and Detragiache (1998) method. They argued that the choice of optimal cut-off thresholds and forecasting periods depends on the degree of risk aversion. In more recent years, Ari (2012) and Sevim et al. (2014) evaluated a currency crisis, but Ari (2012) did not identify the optimal threshold. Conversely, Li et al. (2015) used the index of S&P 500 futures (SP) and the S&P options (SPX) to forecast currency crises. Candelon et al. (2012) evaluated several methods for selecting the absolute optimal cut-off points, concluding that the credit scoring approach and accuracy measures are better than the noise-to-signal ratio. For this reason, we need to identify an effective method to identify the optimal cut-off threshold. The motivation of this paper was to fill the gaps in the current models, by developing a new EWS model, taking macroeconomic and institution indicators into account. The purpose of this paper is to identify the contributing indicators of EWS models to predict currency crises in emerging markets and suggest some policy implications to prevent potential currency crises in future. We hypothesized that there is a positive relationship with the probability of forecasting currency crisis with reserve loss, import growth, real exchange rate growth, and short-term debt/reserve. Similarly, export growth, GDP growth, government stability, corruption, law and order, and external and internal conflicts have the expected negative signs with the dependent variable. This study seeks to identify an optimal cut-off threshold and apply the model to measure and forecast currency crises in any country. Most previous studies focus on identifying economically significant indicators, where the key problem was the ability of those indicators to predict a crisis. We aim to fill the research gap, by identifying key macroeconomic and institution indicators to forecast such currency crises in emerging countries. Determining the optimal threshold is vital, yet many studies use random threshold points, which raises the question of the results' validity. We apply an optimal cut-off threshold identification method, by minimizing two common error types: missing and false alarms. If the chosen cut-off point is too high, fewer currency crises are detected, and missing alarms decrease. Conversely, if the chosen cut-off point is too low, it gives more signals relating to crises; however, false alarms increase.

The contribution of this study includes a detailed analysis of currency crises in emerging countries. Our findings identify significant new variables to explain upcoming crises in sample nations. We use monthly data (from January 1992 to March 2011) from five emerging Asian countries: Indonesia, Malaysia, the Philippines, Thailand, and Turkey and four Latin American emerging countries: Argentina, Brazil, Colombia, and Mexico. We identify a very strong signal of currency crises using the historical data. Employing

logit regression analysis, we find that the macroeconomic and institutional variables are valuable indicators for forecasting currency crises. We find that the real exchange rate, export growth, import growth, current account surplus/GDP, and short-term debt/reserves display the correct signs, as expected, and are statistically significant. We further observe that law and order and external conflict produce the expected signs and are statistically significant. Our contribution further includes developing and enhancing EWS models, by incorporating seven macroeconomic and five institutional variables to cover both sources of crisis indicators. We apply the logit technique to develop the EWS model and credit-scoring approach to identify the cut-off threshold, contrary to the setting of a random threshold point applied by many previous studies. Our model can separate crisis periods and tranquil periods, through its optimal cut-off threshold and minimize the signal errors. Finally, the outcomes of this study will guide policymakers in obtaining better and more precise signals, to employ pre-emptive actions for managing currency crises in future.

The remainder of the paper is structured as follows: Section 2 provides a literature review, Section 3 outlines the materials, methods, and hypotheses development, while Section 4 describes the results of the study and Section 5 presents robustness checks. Finally, Section 6 concludes the paper with policy implications.

## 2. Literature Review

A currency crisis is a situation where speculative attacks on the currency lead to exchange rate depreciation or force the government to protect its currency by increasing the real interest rate or selling foreign reserves. In this empirical study, we considered a currency crisis as a sudden and steep decline in the value of a nation's currency, which causes negative flow throughout the economy. Frankel and Rose (1996) define crises as when the exchange rate depreciation changes greater than 25%, and there is at least 10% change compared to the previous depreciation. Empirical studies identify currency crises by measuring changes in the exchange rate (Frankel and Rose 1996); the exchange rate and international reserves (Kaminsky et al. 1998); the nominal exchange rate; international reserves; and the interest rate (Eichengreen et al. 1995, 1996; Bussiere and Fratzscher 2006); and the real exchange rate, international reserves, and the real interest rate (Bussiere and Fratzscher (2006). A good number of empirical studies on the identification and measurement of currency crises have been published. However, the studies vary in the following aspects: countries that are investigated, modelling approach, variables used, methodology, and estimation techniques, etc.

Eichengreen et al. (1995) employed probit model analysis on quarterly data from 1993 to 1995, while Frankel and Rose (1996) utilized annual data from 1971 to 1992. The studies categorized 16 variables into four groups, namely foreign, macroeconomic, external, and capital inflow, to identify the following: how crises occur as a result of low FDI inflows; low foreign reserves; high domestic credit growth; interest rates; and the real exchange rate overvaluation. However, the Frankel and Rose (1996) model was severely criticized by Berg and Pattillo (1999c) for its poor ability to forecast a crisis accurately.

By employing monthly data for the period 1970 to 1995, Kaminsky et al. (1998) propose an EWS model, namely the KLR model, to predict currency crises. This model was effective in predicting the 1997 currency crisis. The study was further partially supported by Berg and Pattillo (1999c) and extended by Edison (2003), who added seven more variables and eight additional countries, in order to apply the signal approach. The study found that reserves, exports, real exchange rate, money multiplier, money supply/reserves, and imports are necessary to forecast a currency crisis. Similarly, Kumar et al. (2003) contend that export growth, real GDP, reserves losses, inflation, contagion, fiscal balance, and commodity prices are important to identify crises. In the same line of research, Esquivel and Larrain (1998) concluded that macroeconomic variables are the most essential components, which can determine crises by employing panel data probit analysis. Alternatively, Burkart and Couder (2002) employed discriminant analysis to forecast crises. The Kamin et al. (2001) model was used to capture the shock persistence by further extending the model

to one year lagging behind of the explanatory variables. It was found that the external balance and external shock variables are more effective than domestic variables.

Berg and Pattillo (1999a) investigated currency crises in Asian countries using the KLR model from different perspectives and identified five variables, namely, real exchange rate, export loss, reserve loss, current account/GDP, and short-term debt/reserves, as making significant contributions to forecasting crises. Similarly, Peltonen (2006) found that the contagion effect, as well as the rise in current account/GDP and budget deficits/GDP and decline in real GDP growth rate, are good indicators for forecasting currency crises. Leblang and Satyanath (2008) discovered that institutional variables, such as government turnover, ununified government, democracy, and political polarization also play a crucial role in identifying crisis. Similarly, Shimpalee and Breuer (2006) reported that unstable government, weak law-and-order, rising corruption, and fixed exchange rate regimes increase the onset of currency crises, whilst modest capital control and bank independence reduce currency volatility. However, the studies demonstrated that the role of variables is context-oriented. For example, Glick and Moreno (1999) explain how variables such as foreign reserves and real domestic credit are key to forecasting crises in Latin American countries, while the real exchange rate is vital for Asian nations.

In a recent study, Boonman et al. (2019) investigated the performance of the EWS model for currency crises based on quarterly data and employed both signal and logit model approaches. Similarly, Berg et al. (2005) used available information, but no actual outcomes were used to forecast. Investigating 54 emerging economies, Reagle and Salvatore (2005) concluded that data revisions are a crucial aspect of a currency crisis and should be recognized and addressed in future research. Frankel and Saravelos (2012) mentioned that predictions being issued in real-time would be impressive, but especially difficult. In their cases, Duca and Peltonenn (2013) and Holopainen and Sarli (2017) used quasi-real-time data to predict financial stress events and banking crises, respectively.

Based on the above discussion, it is evident that the importance of EWS has been acknowledged in the available literature, but there is a lack of consistency. In this study, we have developed a comprehensive and robust EWS model, by addressing the issues indicated above to fill the gap in our knowledge. Additionally, we have considered a long period and a higher frequency of monthly data. Therefore, our time sample includes various crises and the subsequent minuscule effects of these crises, including the GFC of 2008. Our proposed EWS model consists of dating the currency crises; determining the leading variables as indicators of crises; applying a robust statistical method to test and measure the impact of those variables; and selecting the best cut-off threshold, to receive a clear signal on currency crises. Therefore, this research model is expected to be useful for policymakers, in receiving warnings about the possibility of a currency crisis.

### 3. Materials and Methods

*3.1. Sample Selection*

The panel data for this study covers the period January 1992 to March 2011 for the sample of, first, five emerging countries in Asia, namely Indonesia, Malaysia, Philippines, Thailand, and Turkey; and, second, four emerging countries in Latin America, namely Argentina, Brazil, Colombia, and Mexico. The data are in monthly frequency. Macroeconomic data were collected from the International Financial Statistic (IFS, CD-ROM, 2011) and World Bank (WB) database. The institutional data were taken from the International Country Risk Guide (ICRG, 2012).

Our proposed EWS model consists of dating the currency crises; determining the leading variables as indicators of crises; applying a robust statistical method to test and measure the effect of those variables; and selecting an optimal cut-off threshold to receive a clear signal on currency crises. These are presented in Figure 1.

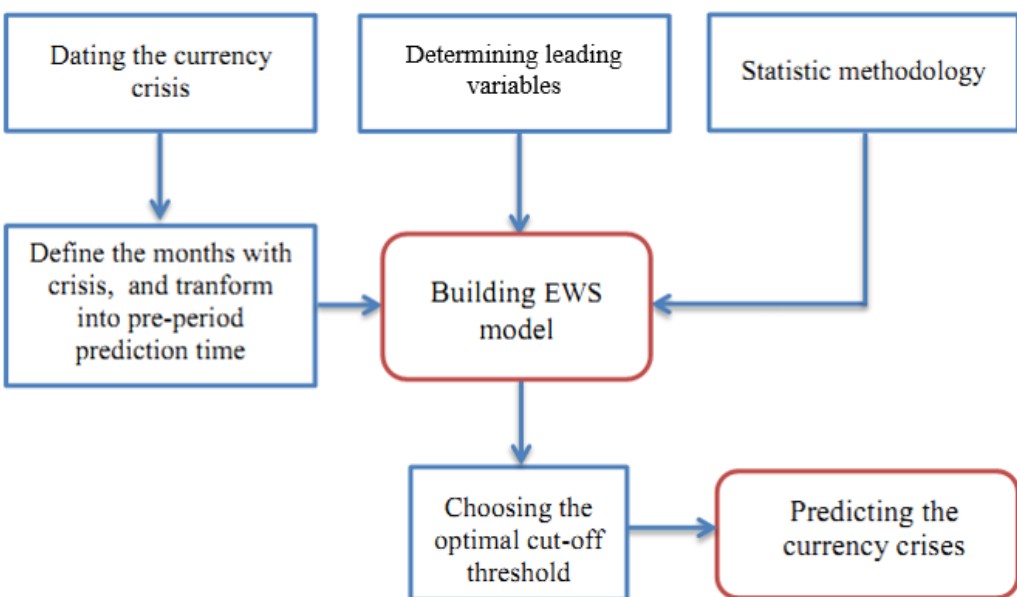

**Figure 1.** The EWS model to predict currency crises.

### 3.2. Variables under Study and Hypotheses Development

Based on the available literature (e.g., Berg and Pattillo 1999b; Kaminsky et al. 1998; Shimpalee and Breuer 2006), we selected seven macroeconomic and five institutional variables that supposedly are positive/negative in association with currency crises. The seven macroeconomic variables include reserve loss, export growth, import growth, real exchange rate growth, current account to GDP ratio, short-term debt to reserve ratio, and GDP growth. The five institutional variables are government stability, corruption, law and order, external conflict, and internal conflict. Higher interest rates lead to a rise in the exchange rate. Additionally, interest rates are predicted to impact exports and imports. That is why interest rates are not included as a predictor in our model. Figure 2 displays the expected relationship between the predictor variables and the currency crisis.

From Figure 2, for those independent variables displaying the expected positive sign with the dependent variable, we hypothesized that there is a positive relationship, with the probability of forecasting currency crises for that particular variable. Similarly, for those independent variables that have the expected negative signs with the dependent variable, we hypothesized that there is a negative relationship, with the probability of forecasting currency crisis for that particular variable.

For instance, a decline in foreign currency reserves is a reliable signal of currency devaluation. When currency collapse occurs, it follows the period typical for maintaining a pegged exchange rate using the foreign reserve. Thus, large falls in foreign reserves cause a high probability of a currency crisis. Moreover, the total value of these foreign reserves is also the variable that illustrates the status of the economy. Consequently, a loss of foreign reserves is a potential indicator of currency crises. The expected sign of this indicator is positive. Therefore, we hypothesized $H_{01}$: Reserve loss positively correlates with the probability of forecasting currency crises.

Alternatively, overvaluation of the exchange rate means that the domestic currency can buy more foreign currency. Hence, exports become more expensive, and demand declines, in turn causing unemployment to rise. Moreover, when the exchange rate is unstable, it leads to speculative attacks on the currency. Therefore, a decrease in export growth can lead to currency crises. The expected sign is negative in the model. Thus, we posited hypothesis $H_{02}$: Export growth negatively correlates with the probability of forecasting currency crises. Similar arguments follow for other independent variables.

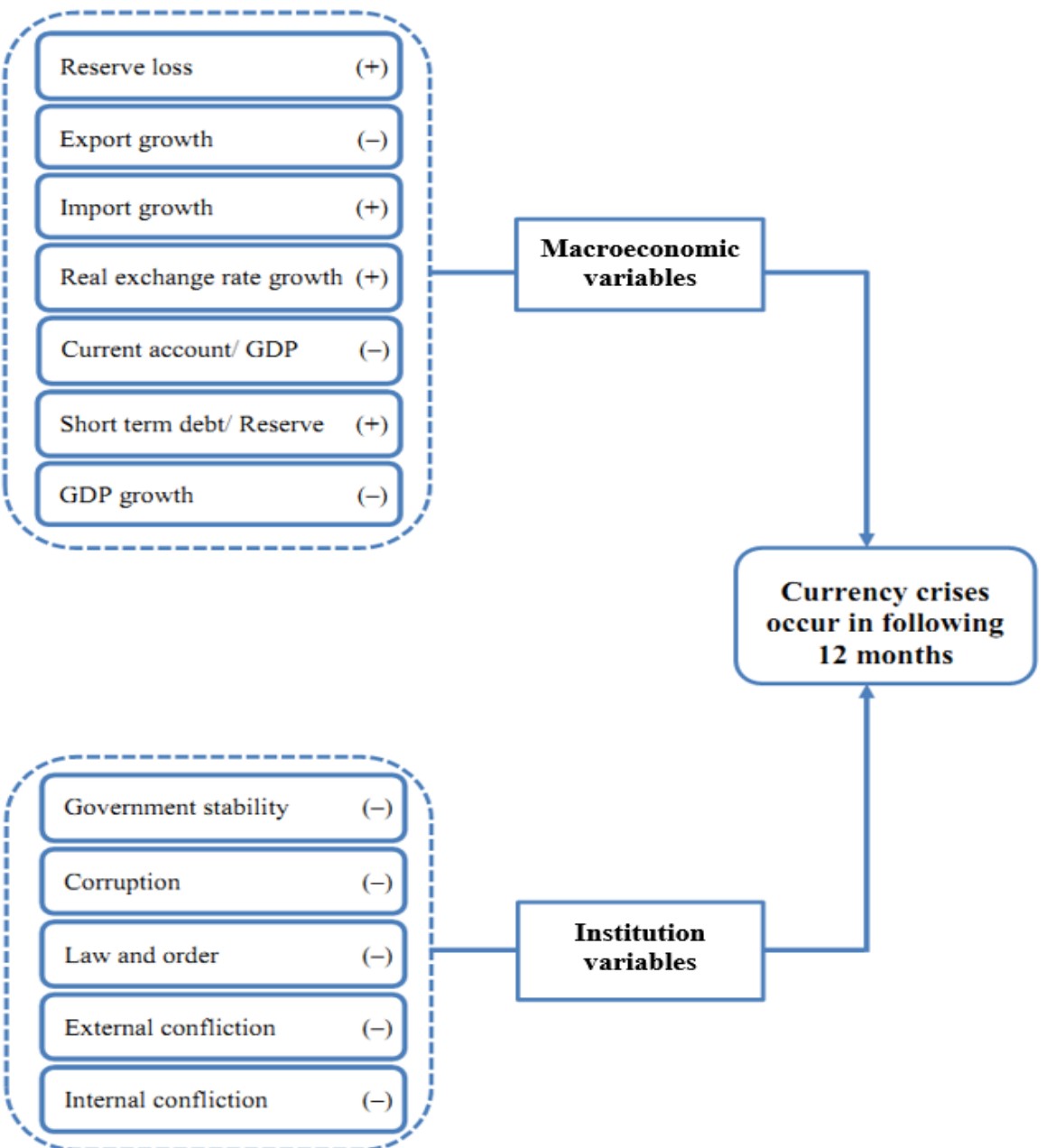

**Figure 2.** Variable selection for hypotheses development.

### 3.3. The EWS Model

This study attempted to examine the probability of currency crises, utilizing the logistic distribution function (Gujarati 2003), as indicated in the following Equation (1):

$$P_{i,t}(Y = 1 | X_{i,t}) = \frac{1}{1 + e^{-(\beta X)}} = \frac{e^{(\beta X)}}{1 + e^{(\beta X)}} \tag{1}$$

where the dependent variable Y is a binary variable (Y = 1 or 0); and the explanatory variables are represented by the following Equation (2):

$$\beta X = \beta_0 + \beta_1 X_{1i,t} + \beta_2 X_{2i,t} + \ldots + \beta_k X_{ki,t} + u_{i,t} \tag{2}$$

Here, $P_{i,t}$ in Equation (1) is the probability of currency crisis occurrence, and $(1 - P_{i,t})$ is the probability of no currency crisis occurrence. $P_{i,t}/(1 - P_{i,t})$ is the odds ratio of the

probability for the currency crisis occurrence to no-occurrence. The ratio is explained by the following Equations (3) and (4):

$$1 - P_{i,t} = 1 - \frac{e^{(\beta X)}}{1 + e^{(\beta X)}} = \frac{1}{1 + e^{(\beta X)}} \tag{3}$$

$$\frac{P_{i,t}}{1 - P_{i,t}} = \frac{e^{(\beta X)}}{1 + e^{(\beta X)}} \times \frac{1 + e^{(\beta X)}}{1} = e^{\beta X} \tag{4}$$

The next step is to arrange the above equation with relevant variables, to formulate the log of odd ratio ($L_{i,t}$), Equation (5) for our EWS model:

$$
\begin{aligned}
L_{i,t} = \ln\left(\frac{P_{i,t}}{1 - P_{i,t}}\right) = \beta X = \beta_0 + \beta_1 RESERVE_{i,t} + \beta_2 EXPORT_{i,t} + \beta_3 IMPORT_{i,t} + \\
\beta_4 RER_{i,t} + \beta_5 CAGDP_{i,t} + \beta_6 STDRES_{i,t} + \beta_7 GDP_{i,t} + \beta_8 GOVERNMENT_{i,t} + \beta_9 LAW_{i,t} + \\
\beta_{10} CORRUPTION_{i,t} + \beta_{11} EXTERNAL_{i,t} + \beta_{12} INTERNAL_{i,t} + u_{i,t}
\end{aligned}
\tag{5}
$$

where dependent variable $L_{i,t}$ stands for 1 = crisis and 0 = no crisis; $\beta_0$ = intercept; explanatory variables $X_{1i,t} = RESERVE_{i,t}$ = Reserve loss; $X_{2i,t} = EXPORT_{i,t}$ = Export growth; $X_{3i,t} = IMPORT_{i,t}$ = Import growth; $X_{4i,t} = RER_{i,t}$ = Real exchange rate growth; $X_{5i,t} = CA\_GDP_{i,t}$ = Current account/GDP; $X_{6i,t} = STD\_RES_{i,t}$ = Short-term debt/Reserve; $X_{7i,t} = GDP_{i,t}$ = GDP growth; $X_{8i,t} = Government_{i,t}$ = Government stability; $X_{9i,t} = LAW_{i,t}$ = Law and Order; $X_{10i,t} = CORRUPTION_{i,t}$ = Corruption (absence of corruption); $X_{11i,t} = EXTERNAL_{i,t}$ = External conflict (absence of external conflict); $X_{12i,t} = INTERNAL_{i,t}$ = Internal conflict (absence of internal conflict) and $u_{i,t}$ = Error term.

The coefficient of the explanatory variables in Equation (2) does not indicate any direct change in the probability of currency crisis occurrence. It, rather, indicates the effect of each explanatory variable on the log of the odds ratio. We can estimate the probability of a currency crisis using Equation (1).

This model attempts to forecast a crisis, and selecting the length of such a forecasting period is always very challenging (Ahsan 2011). A shorter period is always favored, because the chosen variables specified in the model tend to show strong symptoms immediately before the crisis date. Policymakers, however, prefer a longer period, to receive early signals and to save the economy from the upcoming crisis. Previous studies have applied different prediction period ranges. This study utilized a 12-month period, since this is considered a standard length of duration to forecast near future currency crises (Bussiere and Fratzscher 2006).

*3.4. Determination of the Optimal Cut-Off Threshold*

The goal of any EWS model is to identify the crisis and non-crisis period, so that policymakers can execute appropriate policies, in both tranquil and crisis periods. Therefore, in this section, we investigate the method to examine the performance of the EWS model, applying an optimal cut-off threshold. The optimal cut-off is the threshold that separates crisis and tranquil periods. It requires translating the alternative parameter values at time t, where the probability of crisis occurrence within period t + k (k is the specific prediction time) is measured by the Equation (1):

Identifying the cut-off point is crucial. Many other studies have applied an arbitrary method to determine such a threshold (Candelon et al. 2012). Setting an inefficient threshold point can trigger inadequate signals and results. If the probabilities of crises are higher than the cut-off point, this indicates a crisis period; if the probabilities of crisis are lower than the cut-off point, then it should be a tranquil period. When the probability of crises is lower than the appointed cut-off point, the model does not deliver a signal, but the crisis still occurs, so such an incident is called a missing alarm (Type I error). Conversely, when the probability of crisis is higher than the appointed cut-off point, the model delivers a signal, but the crisis does not occur. Such an incident is called a false alarm (Type II error).

If a higher cut-off point is appointed, the model issues more missing alarms and less false alarms. In contrast, if a lower cut-off point is appointed, the model issues less missing alarms and more false alarms. Therefore, finding the optimal threshold is a trade-off between missing and false alarms.

There are a few methods available to select the cut-off threshold. First, to set up a few predefined alternative cut-off points, such as 0.5, 0.25, or 0.1, and then to identify the cut-off point will provide the best signal of a crisis. Policymakers can then decide the useful threshold, guided by these results and indicators (Bussiere and Fratzscher 2006). The arbitrary cut-off method is not very effective, as it does not take Type I and Type II errors into consideration. Second, the noise-to-signal (NSR) ratio, applied by Kaminsky et al. (1998), is a method that selects the optimal cut-off threshold at a point minimizing NSR. The NSR ratio calculates the ratio of false alarms to good signals by considering Type II error and omitting Type I error. Some other methods consider both errors to identify the optimal cut-off threshold, such as the accuracy measures suggested by Lambert and Lipkovich (2008). This method maximizes the correct crisis identification and minimizes the false and missing alarms. The alternative method is the credit-scoring approach suggested by the Basel Committee on Banking Supervision (2005). This method minimizes the absolute value difference between crisis occurrence and no-crisis occurrence; therefore, we apply the credit-scoring approach in this study and explain it in detail below.

The credit-scoring approach applies sensitivity and specificity to identify the cut-off threshold. A simple presentation of the cut-off point selection method with sensitivity and specificity is explained in a matrix by Kaminsky et al. (1998), as follows:

When signal issued for crisis period, Sensitivity $= \frac{A}{A+C}$ and non- crisis period,

$$\text{False alarm} = \frac{B}{B+D}$$

When No signal issued for crisis period, Missing alarm $= \frac{C}{A+C}$ and non- crisis period,

$$\text{Specificity} = \frac{D}{B+D}$$

Here,

A is the number of good signals for a correctly signaled crisis,
B is the number of false alarms, when there is a signal, but no crisis occurs
C is the number of missing alarms, when there is a signal, and a crisis occurs
D is the number of no signals and no crisis occurs

Sensitivity (Se) represents the proportion of correctly identified crises, and specificity (Sp) is the proportion of correctly identified no-crisis occurrences. The rationale of this method is to establish the optimal cut-off threshold (c) at the intersection point of sensitivity and specificity. The sensitivity is downward sloped because, at the high cut-off point, the number of crisis signals issued is low; thus, the percentage of correctly detected crises is low. Conversely, the specification is upward sloped, and due to the high cut-off point, the number of no-crisis signals issued is high. Consequently, the percentage of no-crises identified correctly is also high. The optimal cut-off point is identified at the intersection of sensitivity and specificity, as c = min $|Se - Sp|$.

## 4. Results and Discussion

### 4.1. Empirical Results

This section includes descriptive statistics (Section 4.1), regression results (Section 4.2), the impact of macroeconomic variables (Section 4.3), impact of institutional variables (Section 4.4), and finally, the selection of optimal threshold cut-off, and forecasting crises (Section 4.5).

Descriptive Statistics

The data presented in Table 1 (Panel A, B, C) portray a general picture of the variables selected in the model. The following discussion covers the basic features of each variable, to obtain a better idea about their role in the EWS model. The mean reserve loss of emerging countries in Asia throughout the entire period is −17.19%, the highest and lowest values are between 53.33% and −374.54% (Table Panel A). However, this drops to −18.89% in the tranquil period (Table Panel B) and reaches −2.81% in the crisis period (Table Panel C). Therefore, we can say that when the reserve loss is around −2.81%, it may be one of the signals of a currency crisis.

**Table 1.** Summary of descriptive statistics.

| Table Panel A: Summary of descriptive statistics of the entire period, 1992–2011 | | | | | |
|---|---|---|---|---|---|
| Variables | Obs | Mean | Std. Dev. | Min | Max |
| RESERVE | 1155 | −17.19 | 29.85 | −374.54 | 53.33 |
| EXPORT | 1155 | 13.14 | 22.28 | −68.43 | 371.37 |
| IMPORT | 1155 | 12.59 | 21.41 | −48 | 79.51 |
| RER | 1155 | 32.51 | 86.20 | −44.87 | 837.01 |
| CAGDP | 1155 | 1.33 | 6.24 | −9.73 | 16.85 |
| STDRES | 1155 | 63.36 | 46.70 | 9.05 | 236.20 |
| GDP | 1155 | 4.55 | 4.25 | −13.13 | 10.00 |
| GOVERNMENT | 1155 | 8.24 | 1.60 | 5.08 | 11.00 |
| CORRUPTION | 1155 | 2.52 | 0.80 | 1.00 | 4.00 |
| LAW | 1155 | 3.59 | 0.98 | 2.00 | 5.00 |
| EXTERNAL | 1155 | 3.40 | 1.22 | 1.42 | 5.00 |
| INTERNAL | 1155 | 8.64 | 1.78 | 4.08 | 12.00 |
| Table Panel B: Summary of descriptive statistics for the tranquil period, 1992–2011 | | | | | |
| Variables | Obs | Mean | Std. Dev. | Min | Max |
| RESERVE | 1033 | −18.89 | 29.47 | −374.54 | 32.96 |
| EXPORT | 1033 | 12.26 | 21.60 | −68.43 | 371.37 |
| IMPORT | 1033 | 12.64 | 21.12 | −46.11 | 79.39 |
| RER | 1033 | 23.05 | 65.79 | −44.87 | 482.29 |
| CAGDP | 1033 | 1.57 | 6.23 | −9.73 | 16.85 |
| STDRES | 1033 | 56.61 | 40.61 | 9.05 | 236.2 |
| GDP | 1033 | 4.95 | 3.63 | −13.13 | 10.00 |
| GOVERNMENT | 1033 | 8.18 | 1.58 | 5.08 | 11.00 |
| CORRUPTION | 1033 | 2.46 | 0.78 | 1.00 | 4.00 |
| LAW | 1033 | 3.52 | 0.98 | 2.00 | 5.00 |
| EXTERNAL | 1033 | 3.35 | 1.21 | 1.42 | 5.00 |
| INTERNAL | 1033 | 8.52 | 1.70 | 4.08 | 12.00 |
| Table Panel C: Summary of descriptive statistics for the crisis period, 1992–2011 | | | | | |
| Variables | Obs | Mean | Std. Dev. | Min | Max |
| RESERVE | 122 | −2.81 | 29.24 | −95.61 | 53.33 |
| EXPORT | 122 | 20.62 | 26.26 | −15.58 | 105.8 |
| IMPORT | 122 | 12.15 | 23.88 | −48 | 79.51 |
| RER | 122 | 112.63 | 163.44 | −2.82 | 837.01 |
| CAGDP | 122 | −0.77 | 5.91 | −8.07 | 13.20 |
| STDRES | 122 | 120.47 | 55.29 | 32.28 | 236.20 |
| GDP | 122 | 1.12 | 6.83 | −13.13 | 10.00 |
| GOVERNMENT | 122 | 8.76 | 1.66 | 5.08 | 10.75 |
| CORRUPTION | 122 | 3.04 | .79 | 1.25 | 4.00 |
| LAW | 122 | 4.24 | .68 | 2.50 | 5.00 |
| EXTERNAL | 122 | 3.88 | 1.21 | 2.00 | 5.00 |
| INTERNAL | 122 | 9.59 | 2.15 | 5.58 | 12.00 |

The mean of export growth was 13.14% for the entire period. It had a similar reading during the tranquil period (12.26%), and 20.62% for a crisis period. The import growth

does not indicate any clear signal, because the reading in each period is nearly the same, at around 12.50%.

The gap between the minimum and maximum values of real exchange rate growth during the entire period is very large, between $-44.87\%$ and $837.01\%$, respectively. The minimum value in both the entire period and tranquil period is $-44.87\%$, but drops to $-2.82\%$ during the crisis period. Therefore, $-2.82\%$ could be considered as a signal of crisis.

The mean value of current account surplus/GDP in the whole period is 1.33%; 1% in the tranquil period, while it is $-0.77\%$ in the crisis period. The gap between the maximum and minimum value is not too large, so this indicator should be observed carefully. Moreover, the value of this indicator also depends on GDP growth, so it should be evaluated along with the latter.

The minimum and maximum value of short-term debt/reserve in the entire period is between 9.05% and 236.2%, respectively. It is the same in the tranquil period, but the minimum value in the crisis period is different (32.28%). It could be concluded that, when the value is between 9.05% and 32.28%, this is a signal of better performance. However, when it crosses 32.28%, it needs to be monitored until it reaches the average values of the crisis period (120.47%).

The gap of the maximum (10%) and minimum ($-13.13\%$) value of GDP growth is the same during all periods. Moreover, the mean value of GDP growth in the entire and tranquil periods is at similar level, at 4.55% and 4.95%, respectively. However, the mean value in the crisis period is very low, at 1.12%. Therefore, when the GDP growth is very low, it could be considered a signal of a crisis.

All the institutional variables have a small gap between the maximum and minimum values. The mean values during the crisis period, however, are slightly higher for all variables. The measuring scale for the index of government stability is 0 to 12. The observed maximum and minimum values of this variable range from 8.18% to 8.8%, respectively. Meanwhile, the external conflict ranges from 3.4% to 3.88%, and the internal conflict ranges from 8.52% to 9.59%, over each period of analysis. In addition to this, corruption and law are measured on a scale ranging from 0 to 6. The values range from 2.46% to 3.04% and 3.52 to 4.24%, respectively. Therefore, it is hard to get a clear signal of crisis using the institutional variables, but rather they should be observed and analyzed whenever there is a change in their values.

In Table 2, there is no multicollinearity between the explanatory variables. The value of variance inflation variable (VIF) for all explanatory variables ranges from 1.14 to 3.04, the average of these is 1.79. The correlation coefficients of the explanatory variables are demonstrated in Table 3, and the level of correlation appears to be satisfactory.

**Table 2.** Multicollinearity between explanatory variables.

| Variables | VIF | SQRT VIF | Tolerance | R-Square |
|---|---|---|---|---|
| RESERVE | 1.24 | 1.11 | 0.8074 | 0.1926 |
| EXPORT | 1.42 | 1.19 | 0.7023 | 0.2977 |
| IMPORT | 1.83 | 1.35 | 0.5471 | 0.4529 |
| RER | 1.70 | 1.30 | 0.5899 | 0.4101 |
| CAGDP | 1.55 | 1.25 | 0.6447 | 0.3553 |
| STDRES | 1.96 | 1.40 | 0.5095 | 0.4905 |
| GDP | 1.76 | 1.33 | 0.5666 | 0.4334 |
| GOVERNMENT | 1.14 | 1.07 | 0.8759 | 0.1241 |
| CORRUPTION | 1.95 | 1.39 | 0.5140 | 0.4860 |
| LAW | 2.44 | 1.56 | 0.4100 | 0.5900 |
| EXTERNAL | 1.42 | 1.19 | 0.7059 | 0.2941 |
| INTERNAL | 3.04 | 1.74 | 0.3288 | 0.6712 |

**Table 3.** Correlation analysis.

| | RESE-RVE | EXP-ORT | IMPO-RT | RER | CAGDP | STD-RES | GDP | GOVER-NMENT | CORRU-PTION | LAW | EXTE-RNAL | INTE-RNAL |
|---|---|---|---|---|---|---|---|---|---|---|---|---|
| RESERVE | 1.0000 | | | | | | | | | | | |
| EXPORT | −0.0263 | 1.0000 | | | | | | | | | | |
| IMPORT | −0.2389 | 0.4783 | 1.0000 | | | | | | | | | |
| RER | 0.0767 | −0.0422 | −0.2024 | 1.0000 | | | | | | | | |
| CAGDP | 0.0957 | −0.1336 | −0.2533 | −0.0546 | 1.0000 | | | | | | | |
| STDRES | −0.0251 | 0.0344 | 0.0740 | 0.3785 | −0.4489 | 1.0000 | | | | | | |
| GDP | −0.2579 | 0.0961 | 0.4676 | −0.4121 | −0.3082 | −0.0149 | 1.0000 | | | | | |
| GOVERNMENT | 0.1443 | 0.1118 | 0.0284 | −0.1283 | 0.1311 | −0.1829 | 0.0360 | 1.0000 | | | | |
| CORRUPTION | −0.1177 | 0.1716 | 0.0771 | 0.0714 | −0.2737 | 0.2607 | 0.1978 | 0.0065 | 1.0000 | | | |
| LAW | 0.0505 | 0.1636 | 0.1094 | 0.0882 | −0.2246 | 0.3886 | 0.0934 | 0.1228 | 0.5478 | 1.0000 | | |
| EXTERNAL | 0.1848 | 0.0647 | −0.0556 | −0.2840 | 0.1514 | −0.1915 | 0.0280 | 0.1655 | 0.1460 | 0.2177 | 1.0000 | |
| INTERNAL | 0.1356 | 0.1818 | 0.1076 | −0.3068 | 0.0855 | −0.1707 | 0.1737 | 0.2641 | 0.4890 | 0.5623 | 0.4843 | 1.0000 |

*4.2. Regression Results*

The definition of currency crisis is a sudden and steep decline in the value of a nation's currency, which causes negative flow throughout the economy. As a result, the central bank may not have sufficient foreign exchange reserves to maintain the country's fixed exchange rate. However, there is no concrete measure of currency crisis. In our model, we have used reserve loss as one of the predictors, but one can argue that reserve loss is both a cause and effect of currency crisis[1]. To verify this point, we constructed an alternative model excluding reserve loss. The models do not show any significant difference, moreover, the estimated correlation between the predicted outcomes from two models is 0.98. Hence, we decided to keep the reserve loss in our final model.

Table 4 states the results of the logit model applying Equation (2) for five Asian countries. There are 1155 observations used in this model. The model has a likelihood ratio (LR) statistic 569.71, which is highly significant ($p < 0.000$). Hence, it can reject the null hypothesis (coefficients of all regressors are simultaneously zero). Our model as a whole is significant, and the twelve variables in it are crucial variables.

**Table 4.** Empirical results of logit regression of the 12-month EWS model.

| Indicators | Coefficient (1) | Std. Err. (2) | *p* Value (3) | dy/dx (4) |
|---|---|---|---|---|
| RESERVE | 0.0171211 | 0.0106439 | 0.108 | $5.55 \times 10^{-6}$ |
| EXPORT | −0.0208022 | 0.0093243 | 0.026 | $-6.74 \times 10^{-6}$ |
| IMPORT | −0.0612803 | 0.012334 | 0.000 | 0.0000199 |
| RER | 0.0128762 | 0.0026582 | 0.000 | $4.17 \times 10^{-6}$ |
| CAGDP | −0.4071764 | 0.0669344 | 0.000 | −0.0001319 |
| STDRES | 0.0689924 | 0.0072239 | 0.000 | 0.0000224 |
| GDP | −0.665357 | 0.0891327 | 0.000 | −0.0002155 |
| GOVERNMENT | 1.695356 | 0.1935787 | 0.000 | 0.0005492 |
| CORRUPTION | 1.116541 | 0.3515389 | 0.001 | 0.0003617 |
| LAW | −2.122407 | 0.4992513 | 0.000 | −0.0006876 |
| EXTERNAL | −0.9823548 | 0.2615287 | 0.000 | −0.0003182 |
| INTERNAL | 2.393928 | 0.2752958 | 0.000 | 0.0007755 |
| CONS | −35.9565 | 3.281402 | 0.000 | |

Number of observations = 1155. LR(12) = 569.714. Prob > LR = 0.000. Hosmer-Lemeshow chi2(8) = 2.49. Prob > chi2 = 0.9623.

This study applied the Hosmer–Lemeshow goodness-of-fit test that computes Pearson chi-square from the contingency of observed and predicted frequencies. A goodness-of-fit test yields a high *p*-value. Table 4 demonstrates the Hosmer–Lemeshow chi2 (8) = 2.49 with *p*-value = 0.9623, and this suggests that our model fits the data.

The specification error test results in Table 5 show that _hatsq (*p* = 0.760) is insignificant, so this means the link test is insignificant. The model also does not have specification error, as _hat is significant. Here, the model only included the relevant variables. Furthermore, Table 4 presents all coefficients of regressors as being individually statistically significant at a 5% level, except RESERVE (*p* = 0.108). Moreover, the coefficient of all regressors has

the correct signal, as expected, except GOVERNMENT, CORRUPTION, and INTERNAL. Furthermore, in order to interpret the effect of explanatory variables on the probability of a currency crisis, we calculated the marginal effect of the variables, presented in Table 4 (column 4), followed by an interpretation in Section 4.3, which follows below.

**Table 5.** Specification error test.

|  | **Coefficient** | **Std. Err** | **z** | ***p* Value** |
|---|---|---|---|---|
| _hat | 1.019015 | 0.1183753 | 8.61 | 0.000 |
| _hatsq | 0.0094695 | 0.0310432 | 0.31 | 0.760 |

### 4.3. Impact of Macroeconomics Variables

Reserves loss (RESERVE): reserve loss has a positive impact on the probability of currency crises. This means that a high reserves loss causes a high probability of currency crisis. The marginal effect of reserve loss is very small (0.0000055). Although it has the expected sign, it is not significant at a 5% level ($p = 0.108$); however, the outcome is consistent with Kaminsky et al. (1998).

Export growth (EXPORT): export growth has statistically significant contribution at a 5% level, with the same sign as expected. This finding supports the key role of this indicator in the EWS model; unlike Kaminsky et al. (1998), and Berg and Pattillo (1999a, 1999b). The marginal effect of export growth is $-6.74 \times 10^{-6}$, meaning that, ceteris paribus, when increased by 1 unit, the instantaneous reduction in the likelihood of currency crisis is 0.00067%.

Import growth (IMPORT): this variable has the same signal as expected and is significant at the 1% level. This means that, when the import growth is high, there is a higher chance of a currency crisis occurring. This indicator is significant in our model and emerges as a new indicator; however, Kaminsky et al. (1998) and Berg and Pattillo (1999a) found it to be insignificant. The marginal effect of import growth is 0.0000199, which means that, ceteris paribus, for each additional unit increase in the import growth, the instantaneous probability of currency crises increase by 0.00199%.

Real exchange rate growth (RER): the marginal effect of this variable is $4.17^{-6}$; which means that, ceteris paribus, for each additional unit increase in the growth of the real exchange rate, when the growth of real exchange rate increases by 1 unit, will cause the probability of currency crises to increase by 0.00000417 unit. This variable is significant at the 1% level and has a positive sign, as expected. This result is consistent with many previous studies, such as Kaminsky et al. (1998), Berg and Pattillo (1999a), Glick and Moreno (1999), and Bussiere and Fratzscher (2006).

Current account surplus/GDP (CAGDP): this variable has a negative sign, as expected, and is significant at a 1% level. This means that the higher the value of this ratio, the lower the probability of currency crises. Apart from that, the value of marginal effect is $-0.0001319$, which means that, ceteris paribus, for every additional unit increases in current account surplus to GDP ratio, the instantaneous chance of currency crises occurrence reduces by 0.01319%. This result is similar to findings documented by Berg and Pattillo (1999b), Glick and Moreno (1999), Kamin et al. (2001), and Bussiere and Fratzscher (2006).

Short-term debt/reserves (STDRES): this variable is significant at a 1% level and has a positive sign, as expected. The marginal effect of this variable is 0.0000224. This means that, ceteris paribus, when the ratio of short-term debt to GDP increases by 1 unit, the instantaneous probability of currency crises will increase by 0.00224%. This result reaffirms it as one of the crucial warning indicators for forecasting currency crises. When the value of this indicator increases, this signals an impending crisis. This result is similar to what Bussiere and Fratzscher (2006), and Leblang and Satyanath (2008) reported.

GDP growth (GDP): GDP growth is negative, as expected, and is significant at a 1% level. This means that when the growth of GDP decreases, the probability of currency crisis occurrence increases. The marginal effect of this variable is −0.0002155, suggesting that, ceteris paribus, raising the GDP growth by 1 unit, the instantaneous probability of currency crises will fall by 0.02155%. Bussiere and Fratzscher (2006), Kumar et al. (2003), and Leblang and Satyanath (2008) found similar results.

### 4.4. Impact of Institutional Variables

Government stability (GOVERNMENT): this is significant at a 1% level and is positive, contrary to our expectations. However, Shimpalee and Breuer (2006) found this variable to be positive, which possibly indicates that government stability may be high due to an autocratic/authoritarian regime being in power, coupled with an absence of democracy. Therefore, this variable is still a relevant indicator affecting the probability of currency crises. Nevertheless, a high stability of government will increase the probability of currency crisis occurrence. Furthermore, the marginal effect of government stability is 0.0005492, suggesting that, ceteris paribus, when the index of government stability increases by 1 unit, the probability of currency crises will increase by 0.0005492 units.

Absence of corruption (CORRUPTION): the marginal effect of this variable is 0.0003617. This means that, ceteris paribus, if the value of corruption increases by 1 unit, the instantaneous probability of currency crises will rise by 0.036%. Although this variable is significant at a 1% level, it does not have the same sign as expected. This can be explained according to Berg and Pattillo (1999b), whereby corruption in emerging countries is more widespread. However, it is not an essential signal to predict currency crises.

Law and order (LAW): this is significant at a 1% level and has the sign as expected. The marginal effect of this variable is −0.0006876, indicating that, ceteris paribus, when the index value of law-and-order increases by 1 unit, the instantaneous probability of currency crises will decrease by 0.069%. This result reaffirmed the impact of law and order on the probability of currency crises, supported by the findings of Shimpalee and Breuer (2006). As such, we can say that there is less likelihood of currency crises in a country with a strong law and order system.

Absence of external conflict (EXTERNAL): this variable was not found to be significant by Shimpalee and Breuer (2006); however, we found it had the correct sign and was also significant at a 1% level. The marginal effect of external conflict states that, ceteris paribus, when the value of external conflict increases by 1 unit, the instantaneous probability of currency crisis occurrence will decline by 0.032%. In general, external conflicts, such as war and foreign pressure, are signals of an impending currency crisis.

Absence of internal conflict (INTERNAL): the marginal effect of this variable is 0.0007755. This means that, ceteris paribus, if the value of absence of internal conflict increases by 1 unit, the instantaneous probability of currency crises will increase by 0.078%. While, Shimpalee and Breuer (2006) found this variable to be insignificant, we found that it was significant at a 1% level, but with the opposite sign. This outcome may be linked to an autocratic government being in power, which may also cause more internal conflict. At the same time, however, such a government can control, intervene in, and suspend the standard market mechanisms.

### 4.5. Selection of the Optimal Cut-Off Threshold and Forecasting Crises

This study applies the credit-scoring approach to identify the optimal cut-off threshold, to evaluate the EWS model's performance.

Table 6 and Figure 3 present the identification of optimal cut-off threshold at 13.27%. This is the intersection point of sensitivity and specificity.

**Table 6.** EWS model performance with different cut-off points.

| Cutoff | Sensitivity (Se) | Specificity (Sp) | Missing Alarm (Type I) | False Alarm (Type II) | Type I + Type II |
|---|---|---|---|---|---|
| >=0.05 | 95.90% | 89.45% | 4.10% | 10.55% | 14.65% |
| >=0.10 | 94.26% | 93.13% | 5.74% | 6.87% | 12.61% |
| >=0.1327 | 94.26% | 93.90% | 5.74% | 6.10% | 11.84% |
| >=0.15 | 92.62% | 94.39% | 7.38% | 5.61% | 12.99% |
| >=0.20 | 90.98% | 95.16% | 9.02% | 4.84% | 13.86% |
| >=0.30 | 86.89% | 96.22% | 13.11% | 3.78% | 16.89% |
| >=0.40 | 81.15% | 97.48% | 18.85% | 2.52% | 21.37% |
| >=0.50 | 77.05% | 98.16% | 22.95% | 1.84% | 24.79% |
| >=0.60 | 72.13% | 98.55% | 27.87% | 1.45% | 29.32% |
| >=0.70 | 68.03% | 99.32% | 31.97% | 0.68% | 32.65% |
| >=0.80 | 58.20% | 99.71% | 41.80% | 0.29% | 42.09% |
| >=0.90 | 47.54% | 100.00% | 52.46% | 0.00% | 52.46% |
| >=0.95 | 34.43% | 100.00% | 65.57% | 0.00% | 65.57% |

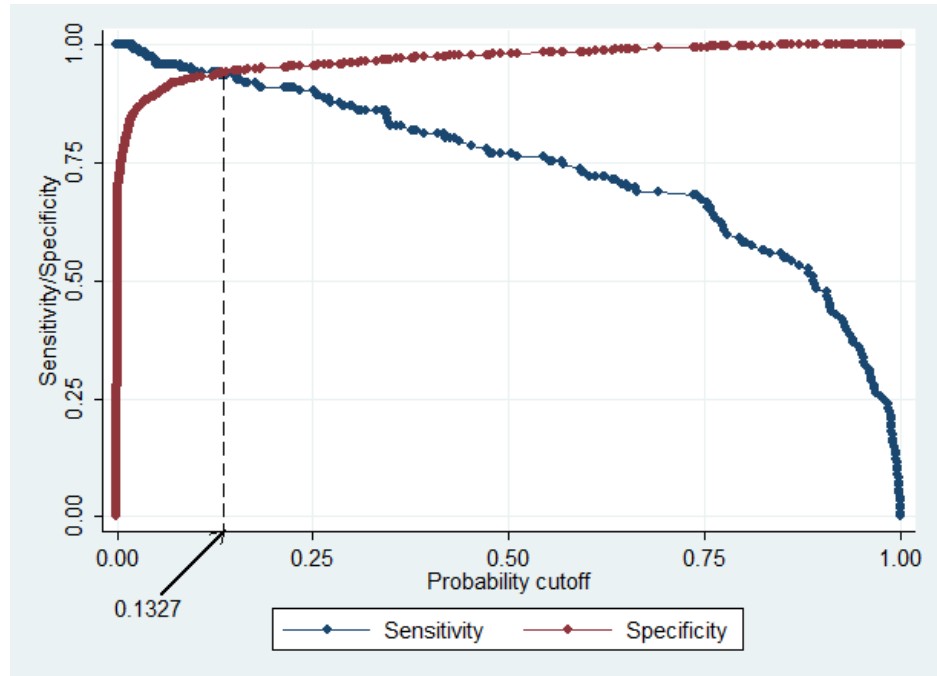

**Figure 3.** Optimal cut-off threshold of the 12-month EWS model in Asian countries.

Table 7 summarizes the EWS prediction of currency crisis applying Equation (2). The probability of crises being predicted correctly (sensitivity) is 94.26%, and the no-crisis prediction (specificity) is 93.90%, false alarm is 6.10%, missing alarm is 5.74%, and observation being called correctly is 93.94%. These results provide evidence that the variables applied are reliable, the sample of countries and time period used is appropriate, and both the Type I and Type II errors are minimal.

The cost of a missing alarm is considered higher than a false alarm, which means the cost of recovery from an actual crisis is higher than the cost of preventing such an event. For this reason, policymakers tend to favor a lower threshold. When the cut-off point is low, there is less probability of missing an alarm and a higher probability of crises occurring. However, total misclassification tends to increase with high false alarms. Moreover, when false alarms increase, policymakers face the risk of the lost cost for pre-emptive actions, such as implementing macroeconomic policies prematurely, to prevent the currency crises, when there is no-crisis occurring (Comelli 2014). In contrast, if the policymakers choose a

higher cut-off point, this curtails the risk of false alarms and the probability of correctly forecasted crises. Thus, the risks with missing alarms is evident.

**Table 7.** Probability of predictability of the 12-month EWS model (cut-off = 13.27%).

|  | Crisis | Non-Crisis | Total |
|---|---|---|---|
| Crisis signal | 115 | 63 | 178 |
| No crisis signal | 7 | 970 | 977 |
| Total | 122 | 1033 | 1155 |
| Number of crises occurrence |  | 115 | |
| % of crises called (sensitivity) |  | 94.26% | |
| % of non-crisis called (specificity) |  | 93.90% | |
| % of false alarm (Type 2 error) |  | 6.10% | |
| % of missing alarm (Type 1 error) |  | 5.74% | |
| % of observation correctly called |  | 93.94% | |

### 4.5.1. Forecasting the Currency Crisis

We input the value of specific time t in the model, to forecast the probability of a currency crisis within the following 12 months. We then compared this probability by applying the optimal cut-off point (13.27%). If the probability of the outcome exceeds the optimal cut-off point, this indicates a signal for currency crises. In contrast, there is no crisis to face if the probability is lower than the optimal cut-off point.

In the following sub-sections, we will use the 12 months preceding the crises to examine our EWS model's ability to forecast the Asian financial crisis in 1997–1998, Turkey's crisis in 1994 and 2001, and the GFC crisis that erupted in 2007.

### 4.5.2. Asian Financial Crisis 1997–1998

We input the value of 12 variables at 1997M1 (January 1997) in our EWS model, and it produced high probabilities such as 95.39%, 77.40%, 99.64%, and 98.97%, indicating an imminent crisis in Indonesia, Malaysia, the Philippines, and Thailand, respectively. There was a low probability (7%) in Turkey (Table 8, column 3). In comparison to the optimal cut-off (13.27%), our EWS model issued signals of crises in Indonesia, Malaysia, the Philippines, and Thailand, but not in Turkey. These results are consistent with the facts, as the Asian financial crisis started in Thailand in July 1997, then spread to Malaysia (July 1997), Indonesia (August 1997), and the Philippines (October 1997), whereas Turkey was unaffected.

**Table 8.** Emerging countries in 1994, 1997, 2001, and 2007.

| Country | Sample in 1994M1 (1) | Crisis in 1994 (2) | Sample in 1997M1 (3) | Crisis in 1997 (4) |
|---|---|---|---|---|
| Indonesia | 0.0086 | No | 0.9539 | 1997M8 |
| Malaysia | 0.0105 | No | 0.7740 | 1997M7 |
| Philippines | 0.0680 | No | 0.9964 | 1997M10 |
| Thailand | 0.0000 | No | 0.9897 | 1997M7 |
| Turkey | 0.4384 | 1994M4 | 0.0007 | No |

| Country | Sample in 2001M1 (5) | Crisis in 2001 (6) | Sample in 2007M1 (7) | Crisis in 2007 (8) |
|---|---|---|---|---|
| Indonesia | 0.0000 | No | 0.0001 | No |
| Malaysia | 0.0114 | No | 0.0000 | No |
| Philippines | 0.0006 | No | 0.0000 | No |
| Thailand | 0.0004 | No | 0.0000 | No |
| Turkey | 0.6188 | 2001M2 | 0.0019 | No |

4.5.3. Turkey's Crisis in 1994 and 2001

Turkey has experienced two crises, in 1994M4 (April 1994) and the other in 2001M2 (February 2001). Column (1) in Table 8 states that the probabilities of crisis in four countries in Asia are low; however, Turkey has a higher probability (43.84%) than the optimal cut-off point; this provides a signal of crisis. In Turkey, the interest rate rose from 75% in December 1993 to 700% in March 1993, followed by currency depreciation by 100%, which pushed the economy into a deep currency crisis in April 1994 (Ali Ari 2012).

The test was repeated in Turkey for the 2001 crisis. Column (5), Table 8 shows that the probabilities for Indonesia, Malaysia, the Philippines, and Thailand are very low. However, Turkey had a higher probability (61.88%) of crises, which is higher than the optimal cut-off threshold, to count it as a valid signal of a crisis to face within the following 12 months. This result is consistent with fact that Turkey faced another crisis in February 2001 (2001M2)

Our EWS model correctly forecast the crises in five Asian countries from 1997 to 1998, and Turkey in 1994 and 2001. Table 8 (column 7) also indicates that the probabilities for the five countries in January 2007 are very low. This shows that, while the other countries were facing the GFC originating in the US, five Asian countries were tranquil by comparison.

An analysis of missing and the false alarms is presented in Table 9. This shows that it missed only one signal in Malaysia (December 1996), five signals in the Philippines (August 1996 to January 1997), one signal in Turkey (December 2000), and no signals were missed in Indonesia and Thailand. However, the number of false alarms was higher in all five countries. While there were three false signals in Indonesia (October to December 1998), five false signals in Malaysia (March 1995, May to June 1995, January 1996, September 1998), 12 false signals in both Philippines (January to April 1996, July to December 1998, May 1999, December 1999) and Thailand (January to June 1996, July to December 1998, and 31 false alarms in Turkey (January 1992 to January 1993, September to December 1994, January 2000, March to December 2001, February to March 2003 and December 2003). Figure 4 illustrates the fit of the actual and predicted values of crises. It shows that all the missing and false signals happened before or after the crises occurred. The false signals prior to a crisis can be explained, in that the governments of these countries implemented policies to prevent the crises, such as selling foreign reserves, increasing the interest rate to maintain the peg exchange rate and stopping currency devaluation. However, such actions did not provide long-lasting benefits, and ultimately Thailand had to float the exchange rate in July 1997. Subsequently, the Thai baht depreciated by 10% and faced yet further devaluations.

**Table 9.** Performance of EWS model in Asian countries with of cut-off = 13.27%.

| Country | Crisis | Non-Crisis | Missing Alarm | False Alarm |
|---------|--------|------------|---------------|-------------|
| Indonesia | 21 | 207 | | 3 |
| Malaysia | 20 | 205 | 1 | 5 |
| Philippines | 19 | 195 | 5 | 12 |
| Thailand | 24 | 195 | | 12 |
| Turkey | 31 | 168 | 1 | 31 |

The false signal issued after the crises may be due to the actions taken during the recovery period, which is called the post-crisis effect by Bussiere and Fratzscher (2006). Our model was not designed to determine the length of the crisis period, rather only to forecast the probability of crisis occurrence. The crisis period in one country could be longer in another country and may be the source of false alarms. Moreover, the model functioned correctly by issuing no additional signals for five Asian countries till the end of 2011.

Overall, it can be concluded that our EWS model correctly signaled the currency crisis in Asia. However, we would like to check the robustness of our model by applying it to an out-of-sample of four emerging countries in Latin America.

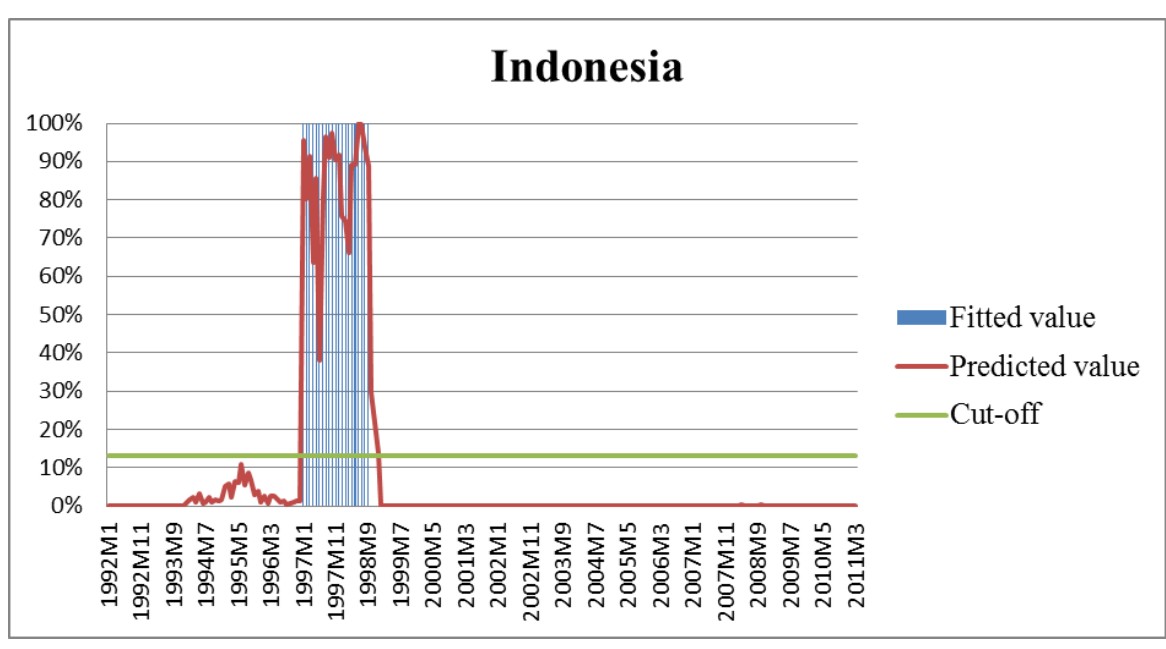

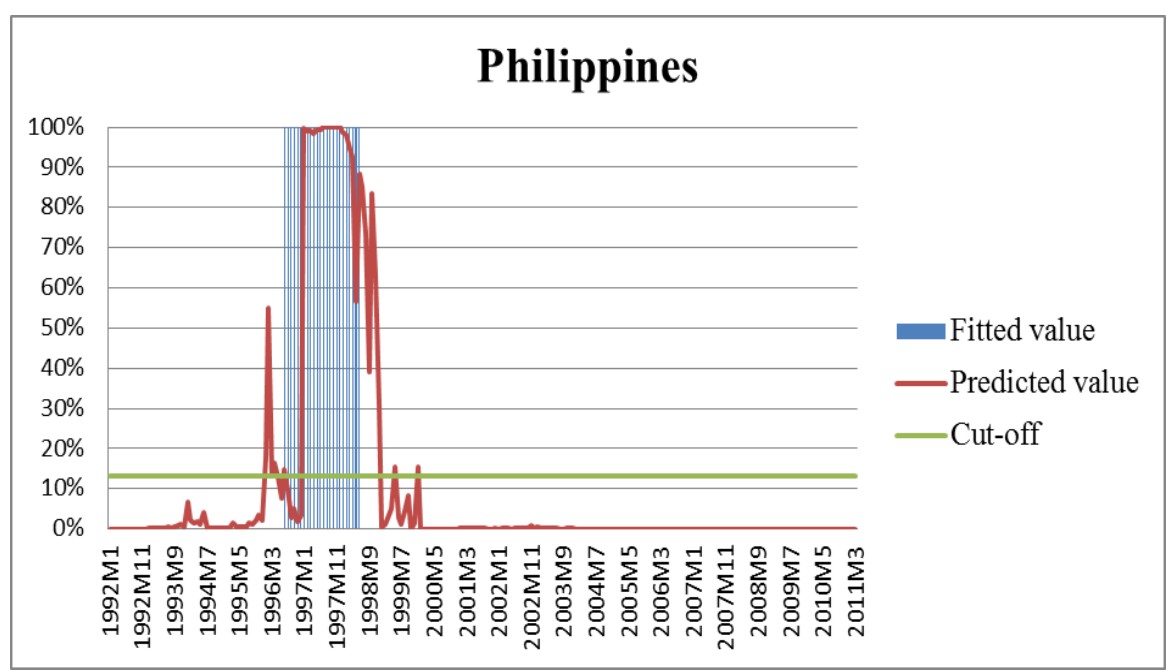

**Figure 4.** *Cont.*

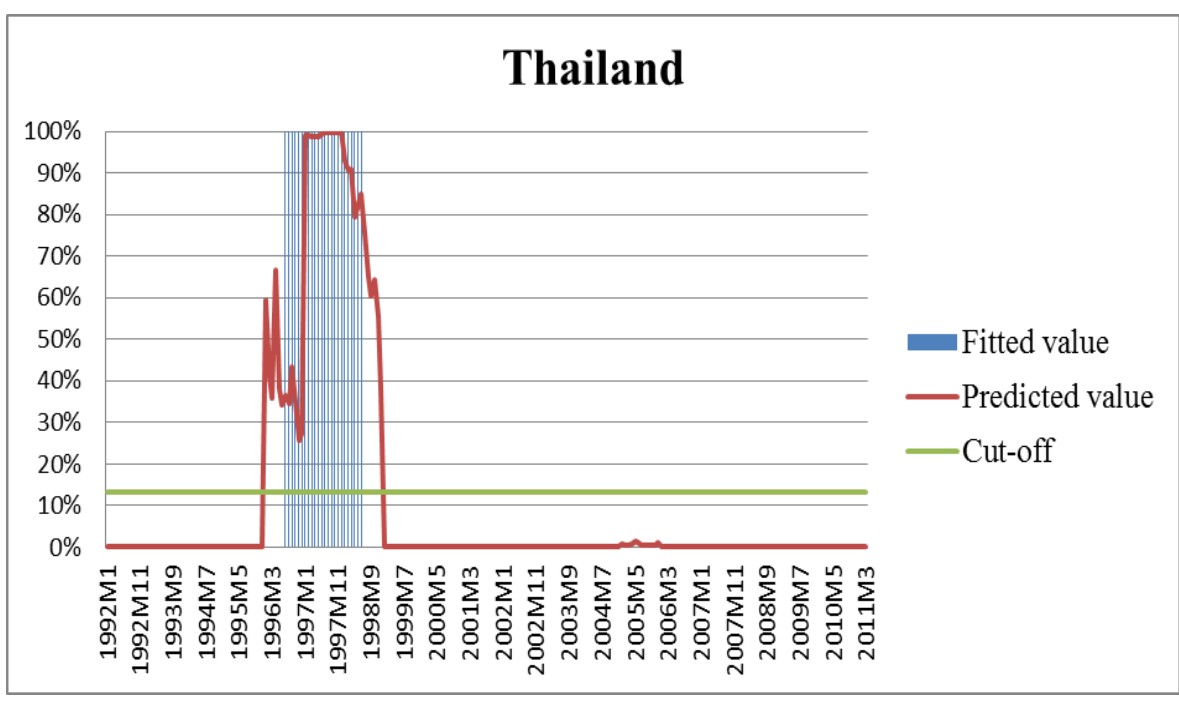

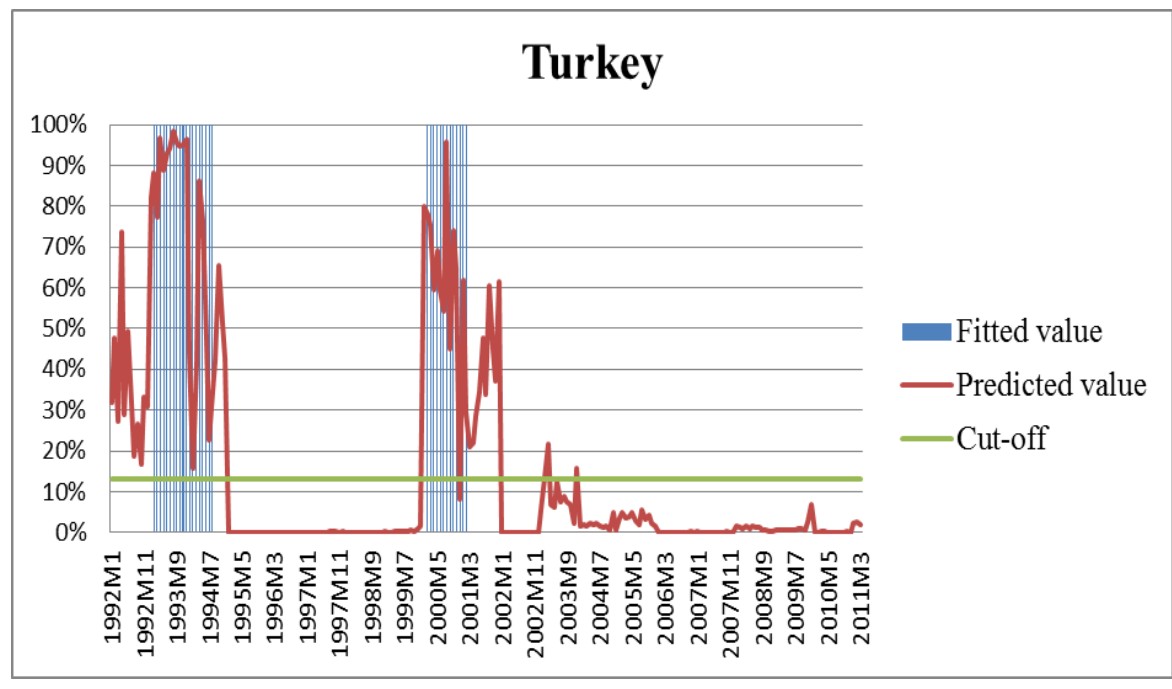

**Figure 4.** *Cont.*

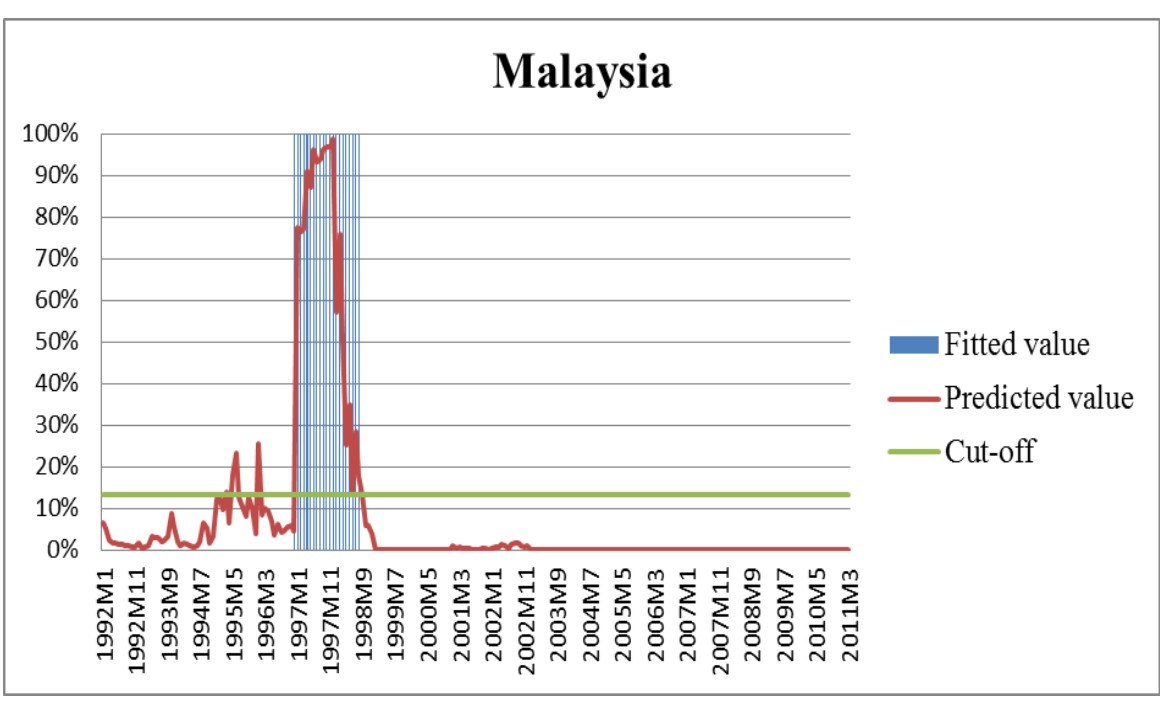

**Figure 4.** Actual and predicted value of five Asian emerging countries.

## 5. Robustness Checks of the EWS Model

### 5.1. Out-of-Sample Test on Latin America

We applied our EWS model to an out-of-sample for four Latin American emerging countries: Argentina, Brazil, Colombia, and Mexico. Table 10 shows that the likelihood ratio (LR) statistic is 442.90 and significant ($p = 0.000$), and Hosmer–Lemeshow chi2 has a $p$-value = 0.9897, which indicates our model's goodness-of-fit.

**Table 10.** Results of EWS model in Latin American countries.

| Variables | Coefficient | Std. Error | $p$-Value | dy/dx |
|---|---|---|---|---|
| RESERVE | 0.0509148 | 0.0114312 | 0.011 | 0.0000319 |
| EXPORT | 0.0243299 | 0.0147422 | 0.099 | 0.0000152 |
| IMPORT | −0.0097348 | 0.0103742 | 0.307 | $-6.10 \times 10^{-6}$ |
| RER | 0.0000538 | 0.0000132 | 0.012 | $3.37 \times 10^{-8}$ |
| CAGDP | −0.3175547 | 0.1117279 | 0.000 | −0.000199 |
| STDRES | 0.0569191 | 0.0103949 | 0.000 | 0.0000357 |
| GDP | −0.0962102 | 0.117412 | 0.370 | −0.0000603 |
| GOVERNMENT | −1.692191 | 0.2108984 | 0.000 | −0.0010604 |
| CORRUPTION | 0.8619664 | 0.4716313 | 0.015 | 0.0005402 |
| LAW | 3.380178 | 0.6085447 | 0.000 | 0.0021182 |
| EXTERNAL | −1.416234 | 0.7054774 | 0.002 | −0.0008875 |
| INTERNAL | −2.507323 | 0.3633715 | 0.000 | −0.0015712 |
| _CONS | 19.22054 | 4.336006 | 0.000 | |

Number of observations = 924. LR(12) = 442.90. Prob > LR = 0.000. Hosmer-Lemeshow chi2(8) = 1.66. Prob > chi2 = 0.9897.

The macroeconomic variables: the reserve loss, real exchange rate growth, current account surplus/GDP, and short-term debt/reserve have the correct sign and remained significant, while the sign changed for export growth and import growth, indicating a new meaning of these variables for Latin American countries. However, the institutional variables have the correct signs and are significant at a 1% level, except for corruption at 5%. Moreover, the government stability changes to a positive sign and is significant at 1%, indicating that the stability of government is crucial and can have different effects on

currency crises compared to Asia. This research suggests that the previously mentioned countries in Latin America should consider having adequate pre-emptive actions to manage internal conflicts to prevent currency crisis.

### 5.2. Selection of the Optimal Cut-Off Threshold and Predict Crises in Latin America

The identification process of optimal cut-off threshold for Latin American emerging countries followed the same method for Asian emerging countries. Figure 5 illustrates the sensitivity and specificity intersected at 12.02%, in order to identify the optimal cut-off threshold for these Latin American emerging countries.

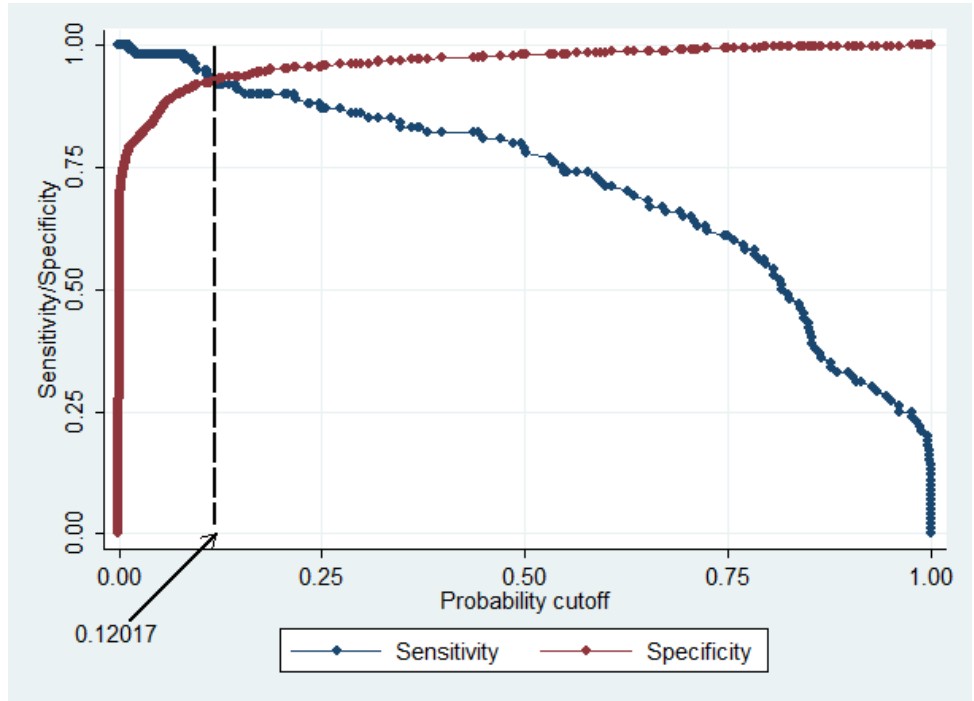

**Figure 5.** Optimal cut-off threshold of the 12-month EWS model in Latin America.

Table 11 presents a summary of the currency crisis forecasting in Latin American emerging countries. It demonstrates that the probability of currency crises is 92.00%, a false alarm is 8.00%, missing alarm is 7.16%, and the correct forecasting of total observations amounts to 92.75%.

**Table 11.** Probability of predictability of the 12-month EWS model (cut-off = 12.02%).

|  | Crisis | Non-Crisis | Total |
|---|---|---|---|
| Signal | 92 | 59 | 151 |
| No signal | 8 | 765 | 773 |
| Total | 100 | 824 | 924 |
| Number of crises that occurred |  | 92 |  |
| % of crises called (sensitivity) |  | 92.00% |  |
| % of non-crisis called (specificity) |  | 92.84% |  |
| % of false alarm (Type 2 error) |  | 8.00% |  |
| % of missing alarm (Type 1 error) |  | 7.16% |  |
| % of observation correctly called |  | 92.75% |  |

Table 12 reports that during the 1992 to 2011 period, our EWS model did not miss any signals in Argentina, and missed just one signal in Mexico, two signals in Brazil, and five signals in Colombia. As in the experiences in Asian emerging countries, the false alarms were higher and issued before or after the good signals for actual crises.

**Table 12.** Performance of EWS model in Latin American countries, cut-off = 12.02%.

| Country | Crisis | Non-Crisis | Missing Alarm | False Alarm |
|---|---|---|---|---|
| Argentina | 20 | 202 | | 9 |
| Brazil | 17 | 208 | 2 | 4 |
| Colombia | 32 | 168 | 5 | 26 |
| Mexico | 23 | 187 | 1 | 20 |

Figure 6 fit the actual and predicted values of four Latin American countries. This indicates that the EWS model detected signals before the actual currency crises in Mexico and Argentina in 1994–1995, and in Argentina in 2001–2002. It also confirms that these countries did not face crises during 2007–2008, and until the end of 2011.

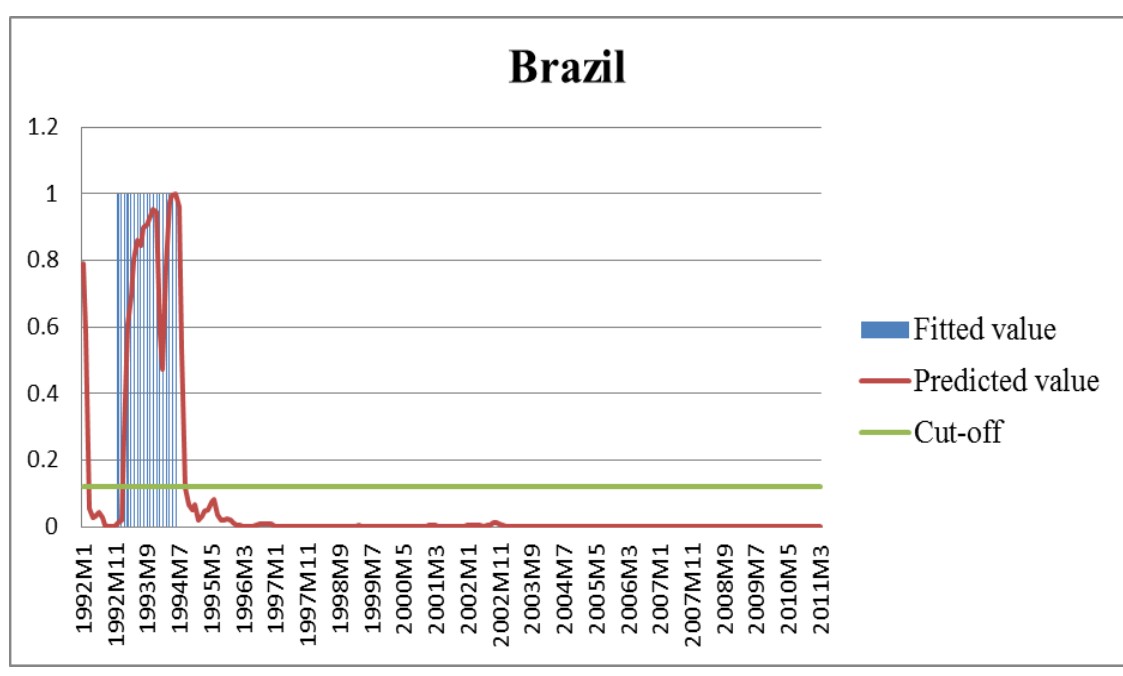

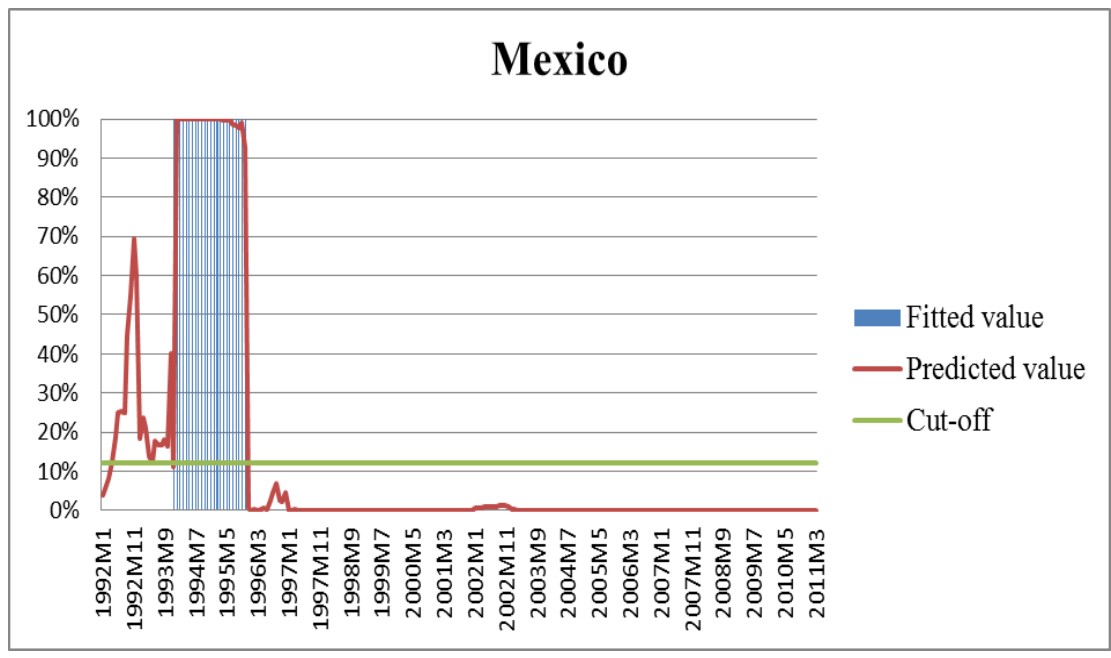

**Figure 6.** *Cont.*

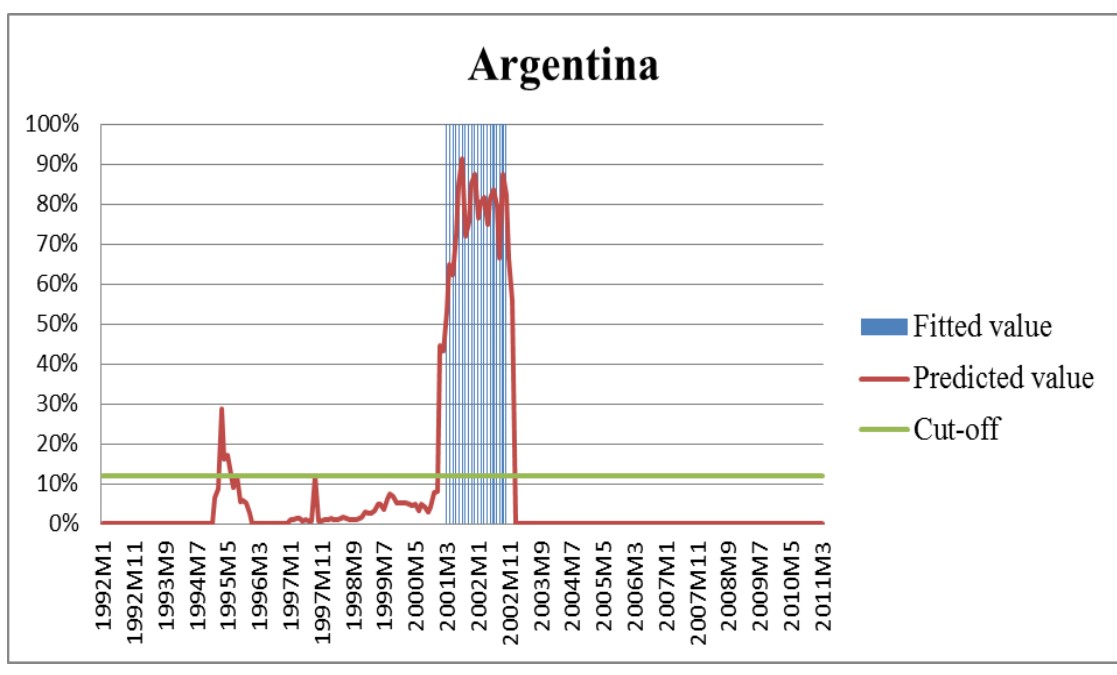

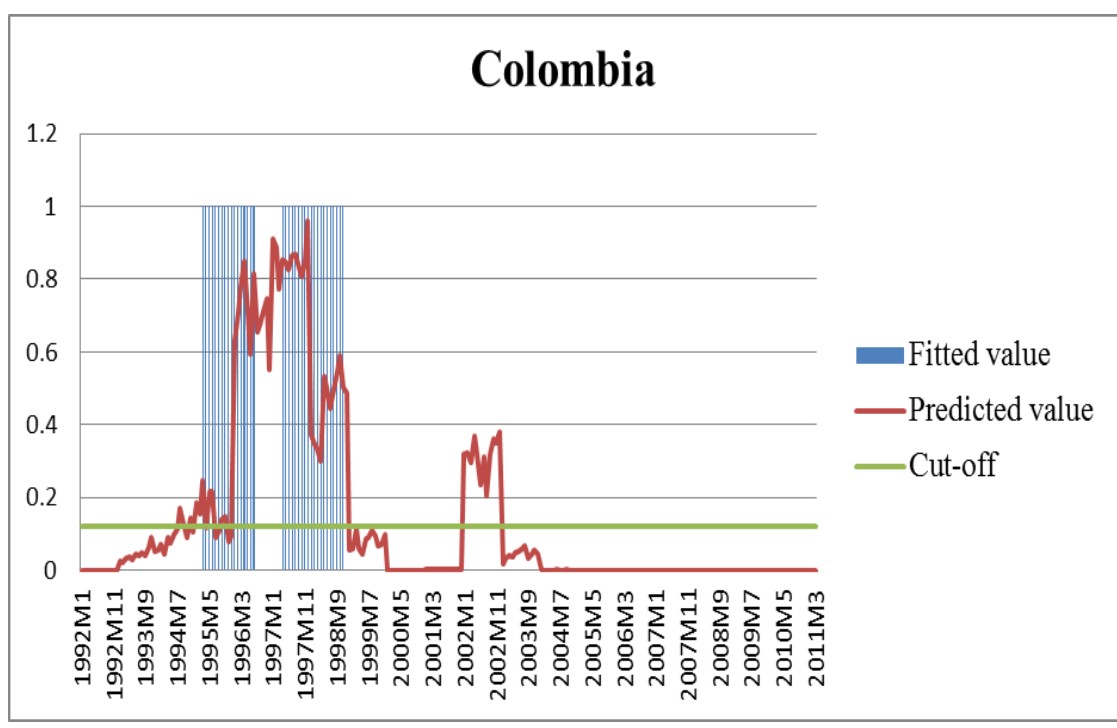

**Figure 6.** Actual and predicted values of four Latin American countries.

The results of out-of-sample Latin American emerging countries confirmed the robustness of our EWS model. It was able to identify and forecast the currency crises in different countries.

## 6. Conclusions

Many countries can suffer repeated currency crises, and in recent years they have caused substantial economic damage to these economies. This study develops a comprehensive EWS model, incorporating all essential components to forecast such currency crises in emerging countries. In addition to developing a robust and comprehensive EWS model, our key contributions include identifying the role of both macroeconomic and institutional

variables, to help track and forecast a currency crisis. The correct interpretation of the movement of such variables can provide valuable information and policy implications. More specifically, we observe that low export growth, current account surplus/GDP, GDP growth, and a high level of the real exchange rate growth, import growth, and short-term debt/reserves, help explain the onset of a currency crisis. In addition to the macroeconomic variables, the institutional variables also contribute to forecasting currency crises. We found that low levels of law and order, coupled with high external conflict will lead to a currency crisis. Interesting findings include high government stability and absence of internal conflict, which may be due to an autocratic government being in power, and ultimately leading to a currency crisis. In terms of macroeconomic variables, our findings are in line with the findings of several other studies, such as Kaminsky et al. (1998), Berg and Pattillo (1999a), Glick and Moreno (1999), Bussiere and Fratzscher (2006), and Leblang and Satyanath (2008). Additionally, in terms of institutional variables, our findings are comparable with Berg and Pattillo (1999b), Shimpalee and Breuer (2006), and so on. However, in the case of the internal conflict factor, our results are in sharp contrast with those of Shimpalee and Breuer (2006).

The key findings indicate that to strengthen the overall macroeconomic performance of a country, to reduce the probability of a currency crisis, a government should take steps to attain higher GDP growth. Our findings further suggest that a country heavily dependent on imports, but with insufficient exports, is likely to face a currency crisis. Excessive dependence on imports also causes current account deficits and opens up paths to a crisis situation. In addition, a country should maintain a stable real exchange rate, to stop speculative attacks on its currency. A trade-off between holding pegged exchange rates and maintaining inflation is necessary, to prevent fluctuations in the real exchange rate. Some countries use short-term foreign debt as an easy solution to manage their liquidity positions; however, such dependence is counterproductive. Short-term repayment conditions coupled with unfavorable exchange rates can cause currency crises. Therefore, it is strongly suggested that short-term foreign debts should be reduced or turned into long-term debts.

We also found that the currency crisis is not only caused by macroeconomic variables but also by institutional variables. In order to improve a country's economy, there should be a strong law and order system in place and conscious efforts made to refrain or reduce conflicts or tensions with other countries. Such systems will promote higher foreign and domestic investment, stimulate economic growth, keep inflation low, and improve current account status, trade deficits, and budget conditions. Our findings also indicate that government stability and the absence of internal conflicts are always desirable. However, this should not be achieved at the expense of threatening democracy and freedom of speech. A trusted electoral system and good governance practices are essential elements to safeguard a country from a currency crisis.

**Author Contributions:** Conceptualization, N.H.T.H., L.T.F., A.A. and K.M.M.K.; methodology, N.H.T.H., L.T.F. and A.A.; software, N.H.T.H. and L.T.F.; validation, A.A., M.S. and L.T.F.; formal analysis, L.T.F.; investigation, N.H.T.H. and L.T.F.; resources, L.T.F.; data curation N.H.T.H. and K.M.M.K.; writing—L.T.F., N.H.T.H., M.S., K.M.M.K. and editing, L.T.F., M.S.; visualization, N.H.T.H. and K.M.M.K.; supervision, L.T.F. and A.A.; project administration, K.M.M.K. and L.T.F.; funding acquisition, L.T.F. All authors have read and agreed to the published version of the manuscript.

**Funding:** This research received no external funding.

**Institutional Review Board Statement:** Not applicable.

**Informed Consent Statement:** Not applicable.

**Data Availability Statement:** Not applicable.

**Conflicts of Interest:** The authors declare no conflict of interest.

## Notes

[1]    We thank one anonymous reviewer for raising this point.

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
