# Peer review of "An Early Warning System for Currency Crises in Emerging Countries"

_jrfm, doi:10.3390/jrfm15040167_

Round 1

Reviewer 1 Report

  1. The structure of the paper presented in lines 108-111 in section 1. Introduction must be correlated with the sections in the paper. In these lines 6 sections are presented, while the paper contains only 5 sections. The paper has not section 2. The literature review mentioned in lines 108-109.
  2. I suggest that in section 1.Introduction to be clearly presented the objectives / hypotheses of the research proposed and carried out in this paper.
  3. I suggest that the conclusions presented in the paper based on the results obtained be compared / correlated with other similar studies that are found in the bibliographic references. If necessary, the authors can complete the list with bibliographic references.

Author Response

Journal of Risk and Financial Management

An early warning system for currency crises in emerging countries

Manuscript ID: jrfm-1602547

We thank the editor for granting us the opportunity to revise and resubmit our paper. We also thank the three anonymous reviewers for providing us with constructive comments and valuable suggestions throughout the review process. We have addressed all comments and concerns raised by the referees. We believe that the paper has improved considerably due to the review comments. Please find our detailed responses to the specific comments of the reviewers below.

Comment of Reviewer 1

  1. The structure of the paper presented in lines 108-111 in section 1. Introduction must be correlated with the sections in the paper. In these lines 6 sections are presented, while the paper contains only 5 sections. The paper has not section 2. The literature review mentioned in lines 108-109.

Response: 

We thank the reviewer for the encouraging and insightful comments. We have taken the reviewer’s suggestion on board and re-structured this section by introducing Section 2 and followed by Section 3,4 and 5 from the earlier version which is presented in lines 125- 129.

  1. I suggest that in section 1. Introduction to be clearly presented the objectives / hypotheses of the research proposed and carried out in this paper.

Response: 

We thank the reviewer for the encouragement and insightful comments made here.

We have worked on the introduction section extensively and included objectives and hypotheses to address your suggestions. In the revised paper, we have noted this in the introduction section from lines 80 to 87 outline the implications of the study for the literature on this topic.

  1. I suggest that the conclusions presented in the paper based on the results obtained be compared / correlated with other similar studies that are found in the bibliographic references. If necessary, the authors can complete the list with bibliographic references.

Response: 

Thank you for flagging this important issue.

We have extensively re-written the entire literature review section along the lines suggested. In our major revision, we have improved and strengthened this section by incorporating new discussions based on the key theoretical perspectives. Additionally, the comparative studies are also referred in the conclusion section (lines 732-737).

Reviewer 2 Report

The contribution of the paper, limited to a sample of emerging countries (ending in 2011), is the explore the role played by institutional variables beyond the macro fundamentals in predicting the currency crises

The following are my comments:

1) Although there is no crystal clear explanation on how the authors identify currency crisis, they focus on  only on exchange rate devaluation. However, they should include also reserve loss to derive an exchange market pressure index which is typically used for currency crisis identification.

Therefore reserve loss should not be used among the predictors

2) It is not clear if the authors carry out of sample forecast evaluation also for Asia, since they refer only to Latin America in section 4 when they refer to this specific prediction evaluation.

3) It is not clear what is exactly the forecast evaluation period

4) There is no mention of QPS and KPS indices of forecasting performance (although they mention them in the literature) which help to summarize via a unique number the forecasting performance

5) There is no benchmark (at least a naive predictor based on the unconditional probability forecast) to evaluate the prediction performance

Author Response

Journal of Risk and Financial Management

An early warning system for currency crises in emerging countries

Manuscript ID: jrfm-1602547

We thank the editor for granting us the opportunity to revise and resubmit our paper. We also thank anonymous reviewer for providing us with constructive comments and valuable suggestions throughout the review process. We have addressed all comments and concerns raised by the referees. We believe that the paper has improved considerably due to the review comments. Please find our detailed responses to the specific comments of the reviewers below.

Comments of Reviewer 2

1) Although there is no crystal clear explanation on how the authors identify currency crisis, they focus on  only on exchange rate devaluation. However, they should include also reserve loss to derive an exchange market pressure index which is typically used for currency crisis identification.

Therefore reserve loss should not be used among the predictors

Response: 

Thank you for raising the issue. A currency crisis is a situation in which serious doubt exists as to whether a country's central bank has sufficient foreign exchange reserves to maintain the country's fixed exchange rate. However, we have defined the currency crisis as the sudden and steep decline in the value of a nation's currency, which causes negative flow throughout the economy. For this reason, we have also considered reserve loss as one of the predictors.

2) It is not clear if the authors carry out of sample forecast evaluation also for Asia, since they refer only to Latin America in section 4 when they refer to this specific prediction evaluation. 

Response: 

Thank you for providing this feedback on the issue. We have discussed sample forecast evaluation also for Asian emerging markets which is included in section 4 and presented in Figure 4.

3) It is not clear what is exactly the forecast evaluation period

Response: 

The developed EWS model can be used to predict currency crises in emerging markets for any period both in-sample and out-sample period. In any given month, it can forecast the probability of a crisis occurring within the following 12 months. For evaluation of the model, the developed EWS model is used for the in-sample period of study for Asian countries and forecasted correctly for the currency crises that occurred in 1997-1998 and in Turkey in 1994 and 2001. In addition, for robustness check of the model, the EWS model is developed for Asian countries only and applied for four Latin American emerging countries: Argentina, Brazil, Colombia, and Mexico for the same period of study. The model is again able to forecast correctly for the currency crises that occurred in these countries. Hence, it is to say that we have performed forecast evaluation for the study period (with both in and out sample countries) and it is possible to apply the model with the identified predictors for any period. 

4) There is no mention of QPS and KPS indices of forecasting performance (although they mention them in the literature) which help to summarize via a unique number the forecasting performance 

Response:

Given the reviewer’s comment, we believe that the mentioned statistics QPS and KPS would not add any additional value. In this paper, we have used the credit-scoring approach that is believed to identify the best recognize crisis period and tranquil period separately as well as minimize the Type I and Type II errors. As an alternative to QPS or KPS, we have considered Area under the ROC curve to measure the performance of the model.  We found that with the cut-off point at 13.27%, our EWS model has the probability of correctly forecasted observations is approximately 93.94% in Asian emerging markets and 92.75% for Latin American emerging markets at the cut-off point of 12.02%.

5) There is no benchmark (at least a naive predictor based on the unconditional probability forecast) to evaluate the prediction performance

Response:

The developed EWS model can be used to predict currency crises in emerging markets for any period both in the sample and out sample period. In any given month, it can forecast the probability of a crisis occurring within the following 12 months.

Reviewer 3 Report

The authors present a model to forecast currency crisis in emerging countries. To do so, they use a logit model. In the following, some suggestions on this work are presented.

The introductory section may include some comments on how recurrent neural networks such as LSTM could help in the modelling proposed in the paper. For example, prediction of variables using this neural network could be used instead of past data.

For example, Oliver Muncharaz (2020) compares different models for time series prediction, in this case, applied to S&P 500 stocks.

Oliver Muncharaz, J. (2020). Comparing classic time series models and the LSTM recurrent neural network: An application to S&P 500 stocks. Finance, Markets and Valuation 6(2), pp. 137–148. https://doi.org/10.46503/ZVBS2781

You can also cite Yuan, J. et al. (2019), whose authors use the neural network LSTM to real-time crash risk prediction. In short, the aim is to make predictions of events (crash traffic, currency crises, etc.) based on a series of variables.

Yuan J, Abdel-Aty M, Gong Y, Cai Q. Real-Time Crash Risk Prediction using Long Short-Term Memory Recurrent Neural Network. Transportation Research Record. 2019;2673(4):314-326. doi:10.1177/0361198119840611

- The use of DOI`s is recommended in all literature.

- Interest rates have not been considered in the selection of variables in the model. It would be convenient to justify the non-inclusion of this variable.

Otherwise, the paper is well developed, with a logical structure and conclusions that support the work carried out.

Author Response

We thank the editor for granting us the opportunity to revise and resubmit our paper. We also thank anonymous reviewer for providing us with constructive comments and valuable suggestions throughout the review process. We have addressed all comments and concerns raised by the referees. We believe that the paper has improved considerably due to the review comments. Please find our detailed responses to the specific comments of the reviewers below.

Comments of Reviewer

1.The authors present a model to forecast currency crisis in emerging countries. To do so, they use a logit model. In the following, some suggestions on this work are presented.

The introductory section may include some comments on how recurrent neural networks such as LSTM could help in the modelling proposed in the paper. For example, prediction of variables using this neural network could be used instead of past data.

For example, Oliver Muncharaz (2020) compares different models for time series prediction, in this case, applied to S&P 500 stocks.

Oliver Muncharaz, J. (2020). Comparing classic time series models and the LSTM recurrent neural network: An application to S&P 500 stocks. Finance, Markets and Valuation 6(2), pp. 137–148. https://doi.org/10.46503/ZVBS2781

You can also cite Yuan, J. et al. (2019), whose authors use the neural network LSTM to real-time crash risk prediction. In short, the aim is to make predictions of events (crash traffic, currency crises, etc.) based on a series of variables.

Yuan J, Abdel-Aty M, Gong Y, Cai Q. Real-Time Crash Risk Prediction using Long Short-Term Memory Recurrent Neural Network. Transportation Research Record. 2019;2673(4):314-326. doi:10.1177/0361198119840611

Response:

Thank you for providing this feedback on the issue. Neural Network modelling is another dimension of analysis for time series prediction. In this paper, we are dealing with a set of panel data. Neural network modelling is beyond the scope of the paper. We hope to explore with respect to this area by applying neural Network modelling in the future study.    

2. The use of DOI`s is recommended in all literature.

Response: 

Thanks for your suggestion here. In our revised manuscript we have included DOI’s in the references list

3. Interest rates have not been considered in the selection of variables in the model. It would be convenient to justify the non-inclusion of this variable.

Response: 

When inflation rises, the purchasing power of the currency is reduced, domestic interest rates increase and borrowing becomes more expensive. Higher interest rates lead to a rise in the exchange rate. Additionally, interest rates are predicted to impact export and import. That is why the interest rate is not included as a predictor in our model.

4. Otherwise, the paper is well developed, with a logical structure and conclusions that support the work carried out.

Response: 

Thanks for your helpful comment and insightful feedback which help us to improve our manuscript.

Round 2

Reviewer 2 Report

In response to Comment 1, the authors do not make any effort in constructing the exchange market pressure index which is what the literature on forecasting currency crises focus on

In response to comment 4), the authors mention of the area under the ROC which is typically used as an indicator of forecasting performance.  Altough the authors mention they've computed this area, I could not find it in the manuscript

In response to comment 5), the author do no make any effort to compare the performance of their model with the one of a naive predictor as I suggested

Author Response

  1. i)  'In response to Comment 1, the authors do not make any effort in constructing the exchange market pressure index which is what the literature on forecasting currency crises focus on'

    I think the literature does not all focus on the exchange market pressure index although it is used a lot.  I think the authors do not need to do entirely what the reviewer suggests but they should explain in some detail in the paper why they have used reserve loss as a predictor when this differs from much of the literature.

Response:

Although the definition of currency crisis is the sudden and steep decline in the value of a nation's currency, which causes negative flow throughout the economy; as a result, the central bank may not have sufficient foreign exchange reserves to maintain the country's fixed exchange rate. However, there is no concrete measure of the currency crisis. In our model, we have used reserve loss as one of the predictors, but one can argue that reserve loss is both a cause and effect of the currency crisis. We thank the anonymous reviewer for raising this point. To verify this point, we constructed an alternative model excluding reserve loss. The models do not show any significant difference in the estimated coefficient.

Moreover, the estimated correlation between the predicted outcomes from the two models is 0.98. Hence, we decided to keep the reserve loss in our final model. This paragraph is included on lines 428-436. 

  1. ii) 'In response to comment 4), the authors mention of the area under the ROC which is typically used as an indicator of forecasting performance.  Although the authors mention they've computed this area, I could not find it in the manuscript'

    The authors should make sure this is clear in the manuscript.

Response:

Thank you very much for this issue. We can confirm that these results regarding forecasting performance using the area under ROC are presented in table 7 for Asian countries and 11 for Latin American countries. The results are explained on page 16 (lines 575 to 580) and on page 20 (lines 696 to 699).

iii) In response to comment 5), the author do no make any effort to compare the performance of their model with the one of a naive predictor as I suggested.

I don't think this is essential

Response:

Thank you very much for your suggestion.